# Structure of the AAA protein Msp1 reveals mechanism of mislocalized membrane protein extraction

Lan Wang[1,2], Alexander Myasnikov[2,3†], Xingjie Pan[4], Peter Walter[1,2]*

[1]Howard Hughes Medical Institute, Chevy Chase, Maryland, United States; [2]Department of Biochemistry and Biophysics, University of California, San Francisco, San Francisco, United States; [3]Centre for Integrative Biology, Department of Integrated Structural Biology, IGBMC, CNRS, Inserm, Université de Strasbourg, Illkirch, France; [4]UCSF/UCB Graduate Program in Bioengineering, University of California, San Francisco, San Francisco, United States

**Abstract** The AAA protein Msp1 extracts mislocalized tail-anchored membrane proteins and targets them for degradation, thus maintaining proper cell organization. How Msp1 selects its substrates and firmly engages them during the energetically unfavorable extraction process remains a mystery. To address this question, we solved cryo-EM structures of Msp1-substrate complexes at near-atomic resolution. Akin to other AAA proteins, Msp1 forms hexameric spirals that translocate substrates through a central pore. A singular hydrophobic substrate recruitment site is exposed at the spiral's seam, which we propose positions the substrate for entry into the pore. There, a tight web of aromatic amino acids grips the substrate in a sequence-promiscuous, hydrophobic milieu. Elements at the intersubunit interfaces coordinate ATP hydrolysis with the subunits' positions in the spiral. We present a comprehensive model of Msp1's mechanism, which follows general architectural principles established for other AAA proteins yet specializes Msp1 for its unique role in membrane protein extraction.

*For correspondence:
peter@walterlab.ucsf.edu

Present address: †Department of Structural Biology, St. Jude Children's Research Hospital, Memphis, United States

Competing interests: The authors declare that no competing interests exist.

## Introduction

ATPases associated with diverse cellular activities (AAA proteins) utilize the energy of ATP hydrolysis to facilitate numerous functions in the cell, such as degrading proteins (*Pickart and Cohen, 2004*), dissolving protein aggregates (*Sanchez and Lindquist, 1990*), or moving proteins across membranes (*Ye et al., 2001*; *Gardner et al., 2018*). Many AAA proteins form homo-oligomers, in which six identical ATPase modules arrange in right-handed spirals surrounding a central pore. Each monomeric subunit (referred to as M1, M2, ..., indicating its position in the spiral) has an N-terminal domain (N-domain) followed by a core-ATPase domain. The N-domain facilitates initial engagement of the substrate. Akin to the six-piston rotary engine, the core-ATPase domains undergo coordinated cycles of ATP hydrolysis. Yet in contrast to a six-piston engine, the AAA molecular motor rebuilds itself during each cycle, with a terminal subunit leaving the spiral from the M6 position and replacing the subunit in the M1 position at the opposite end. The resulting conformational changes result in treadmilling of the spiral along its substrate, which produces a power stroke that drives an unfolded polypeptide chain through the central pore (*Wendler et al., 2012*; *Deville et al., 2017*; *Bodnar and Rapoport, 2017*; *Hinnerwisch et al., 2005*; *Martin et al., 2008*; *Yang et al., 2015*). Pore loops projecting from the ATPases engage with the polypeptide and with each M6→M1 conversion cycle translocate it in steps of two amino acids as the spiral crawls along the substrate (*Gates et al., 2017*; *Monroe et al., 2017*; *Puchades et al., 2017*; *de la Peña et al., 2018*; *Dong et al., 2019*).

Constrained by the ATPase fold, three loops from each subunit extend into the pore. Pore-loop one is well-conserved across the AAA protein family, with a signature motif of one aromatic amino acid followed by a hydrophobic one (*Hanson and Whiteheart, 2005*). The aromatic amino acids directly intercalate between the substrate's amino acid side chains and are arranged as a spiral staircase around the substrate. Pore-loop two is more variable across the family. It protrudes into the central pore, but is either disordered in the existing AAA protein structures, or, as in the recent structures of Yme1 (*Puchades et al., 2017*) and Rix7 (*Lo et al., 2019*), disengaged from the substrate. Hence, although there are indications of pore-loop 2's importance in substrate threading, it has remained unclear whether or not it does so by forming direct contacts. Pore-loop three is short and, likewise, does not contact the substrate directly.

Despite these common architectural features, AAA proteins comprise a diverse superfamily. Each AAA protein harbors unique structural features apparently suited to its particular biological purposes. How these specialized features ensure or contribute binding of the correct substrate in the correct cellular location, or couple ATP hydrolysis to peptide unfolding is largely unknown.

The AAA protein Msp1 (in yeast; ATAD1 in mammals) extracts the tail-anchored (TA) membrane proteins that have failed to be correctly inserted into the ER membrane (*Okreglak and Walter, 2014*; *Chen et al., 2014*). TA proteins comprise an important class of transmembrane proteins. Many of them perform important functions in various processes, including peroxisome biogenesis (Pex15), membrane fusion (SNARE proteins), and apoptosis (bcl-2 family proteins). Many TA proteins (including SNARE proteins) are post-translationally targeted to and integrated into the endoplasmic reticulum (ER) membrane by the GET pathway (TRC40 pathway in mammals) (*Borgese and Fasana, 2011*). TA proteins that escape this reaction are mistargeted to the mitochondrial outer membrane (MOM), necessitating their removal. This proofreading function is performed by Msp1/ATAD1, which extracts the mistargeted TA proteins from the MOM to facilitate their subsequent proteasomal degradation (*Dederer et al., 2019*; *Matsumoto et al., 2019*). Accordingly, deletion of Msp1 or GET pathway components leads to mislocalization of TA proteins to the MOM, and deletion of both to severe synthetic growth defects. In vitro reconstitution experiments showed directly that Msp1 is sufficient to extract TA proteins, which confirmed Msp1's proposed role as a membrane protein dislocase (*Wohlever et al., 2017*). Msp1 has also been implied in the clearance of mitochondrial precursor proteins that are stuck in the MOM import machinery, indicating that its role extends beyond extracting mislocalized TA proteins (*Weidberg and Amon, 2018*).

ATAD1 has likewise acquired an additional role beyond protein quality control. In neurons, it mediates the internalization of AMPA receptors required for synaptic plasticity during long-term depression (*Zhang et al., 2011*). It acts to disassemble AMPA receptor-GRIP1 complexes, freeing AMPA receptors from their scaffolding so that they can be endocytosed. Mice with *ATAD1* deletions die from a seizure-like syndrome, caused by an excess of surface-expressed AMPA receptors, in agreement with ATAD1's crucial role as a regulator of AMPA receptor trafficking.

Among the members of the AAA protein family, Msp1 clusters with spastin, katanin, fidgetin, and Vps4 in one of the six subfamilies called the 'meiotic clade' (MC) (*Iyer et al., 2004*; *Erzberger and Berger, 2006*; *Frickey and Lupas, 2004*). AAA$_{MC}$ proteins share many similar structural features that differentiate them from other AAA protein clades. One feature lies in their pore-loop two sequences, which for AAA$_{MC}$ proteins show strong sequence similarity. Mutations in Msp1, spastin, and katanin pore-loop two lead to significant decrease in their activity (*Wohlever et al., 2017*; *Roll-Mecak and Vale, 2008*; *Shin et al., 2019*; *Zehr et al., 2020*), suggesting the importance of this loop. Recent structures of katanin (*Zehr et al., 2020*) and spastin (*Sandate et al., 2019*) both showed that a positively charged pore-loop two contacts the side chain of the glutamate in the polyglutamate tail of β-tubulin, presumably conferring substrate specificity and neutralizing the charges in the central pore. By contrast, the structure of Vps4-substrate complex contains an ordered pore-loop 2 (*Han et al., 2017*), yet as in many other AAA proteins, Vps4's pore-loop two does not contact the substrate. Msp1's pore-loop two closely resembles those of katanin and spastin, but instead of binding to negatively charged peptides, it extracts hydrophobic membrane proteins. If and how Msp1's pore-loop two contacts the substrate remained unknown.

Another common feature among the AAA$_{MC}$ proteins lies in sequences, referred to as intersubunit signaling (ISS) motifs (*Augustin et al., 2009*), ISS motifs transmit information regarding the adenosine nucleotide-bound state between adjacent subunits and synchronize ATP hydrolysis with pore-loop movement through allosteric conformational changes. The ISS motif found in some AAA

proteins, such as Yme1 and proteasome subunits, contains a crucial phenylalanine in a conserved DGF tripeptide (*Figure 1—figure supplement 1*) (*Puchades et al., 2017*), whereas the ISS in AAA$_{MC}$ proteins lacks this amino acid and instead contain short insertions, indicating that intersubunit signaling must utilize a different mechanism than previously described.

Despite pronounced sequence similarity, Msp1 has distinct features that differentiate it from other AAA$_{MC}$ proteins. One of those features lies in its N-domain. Katanin, spastin and Vps4 contain structurally related domains that recruit the proteins to their substrates (microtubules and ESCRT III complexes, respectively [*Rampello and Glynn, 2017*; *Rigden et al., 2009*; *Monroe and Hill, 2016*; *Sun et al., 2017*; *Su et al., 2017*]). By contrast, in Msp1 the most N-terminal region of the N-domain

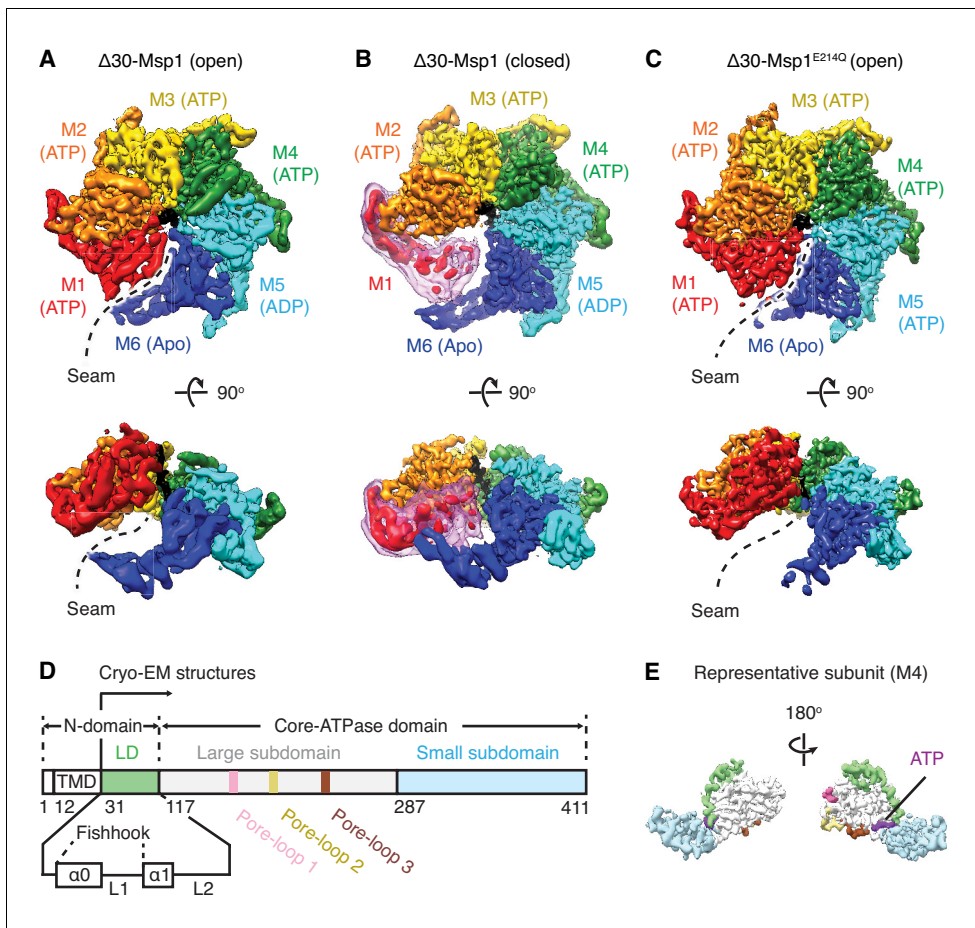

**Figure 1.** Architecture of the Msp1-substrate complexes. (A to C) Final reconstructions of Δ30-Msp1 (open), Δ30-Msp1 (closed) and Δ30-Msp1$^{E214Q}$ complexes shown in top and side views. Each subunit (M1 to M6) is assigned a distinct color, and the substrate is shown in black. The spiral seams of the two open conformations (panels A and C) are denoted with dashed lines. In (B), the map for the mobile subunit M1 is depicted in two thresholds: in red is σ = 5.3 (same to the rest five subunits) and in light pink is σ = 2.5. (D) Schematic of individual domains and structural elements of Msp1. The numbers are based on the *C. thermophilum* Msp1. (E) A representative Msp1 subunit (M4) with domains and structural elements colored according to (D). ATP is shown in purple.

The online version of this article includes the following figure supplement(s) for figure 1:

**Figure supplement 1.** Sequence comparison of AAA$_{MC}$ ATPases to the mitochondrial AAA proteases suggests structural similarity within the meiotic clade.

**Figure supplement 2.** Primary sequence alignment of Msp1 homologs showing conserved structural elements.

**Figure supplement 3.** The SEC traces of the Δ30-Msp1 and the Δ30-Msp1$^{E214Q}$ proteins.

**Figure supplement 4.** Data analysis scheme of the Δ30-Msp1 structures.

**Figure supplement 5.** Data analysis flow for the Δ30-Msp1$^{E214Q}$ structure.

**Figure supplement 6.** Structure of the larger oligomer shows potential steric clash between the additional subunit and the TMD of existing subunits.

is replaced by a transmembrane helix (*Figure 1D*). The transmembrane helix is thought to anchor Msp1 to intracellular membranes, such as the MOM. The transmembrane domain is followed by a linker domain (LD), resembling the most C-terminal part of the N-domain also seen in katanin. Interestingly, the Msp1 LD interacts with and can be crosslinked to hydrophobic patches in substrate proteins, suggesting that it functions to confer specificity on substrate selection. Mutating amino acids in Msp1 LD reduced substrate binding, increased the level of mislocalized TA proteins in cells, and caused severe synthetic growth defects in *get3*-deleted cells. The hydrophobic patch on the substrate was likewise required for recognition and removal by Msp1, as mutating hydrophobic amino acids in this patch to alanine abolished substrate interaction with Msp1. Finally, insertion of a hydrophobic patch into a nonsubstrate (Gem1) rendered it susceptible to membrane extraction by Msp1 (*Li et al., 2019*).

Because hydrophobic patches are expected to be covered by the interaction partner in properly targeted TA proteins, selection of hydrophobic patches provides an intuitive explanation of how Msp1 distinguishes correctly from incorrectly targeted TA proteins, where, in the latter case, cognate interaction partners would not exist. However, how, mechanistically, the LD and perhaps other elements in Msp1 contribute to substrate recognition remains unknown.

A previously reported crystal structure at 2.6 Å of monomeric Msp1 allowed modeling of the monomer into the AAA protein p97 hexamer (*Wohlever et al., 2017*). The analyses provided predictive information regarding intersubunit contacts, yet could not address the mechanistic questions raised above. Here, we present a collection of three high-resolution Msp1 solution-state structures (an ATP hydrolysis-arrested Msp1 mutant and two conformational states of wildtype Msp1) that address these outstanding questions and begin to explain how particular features, shared by AAA$_{MC}$ proteins adapt it to its particular roles in protein quality control and beyond.

## Results

### Msp1-substrate complexes adopt open and closed spiral conformations

To obtain a homogeneous sample suitable for structural studies, we expressed the cytosolic domain of Msp1 lacking its 30 amino acid N-terminal membrane anchor (Δ30-Msp1) from the thermophilic yeast *Chaetomium thermophilum* (*Figure 1—figure supplement 2*). In addition, we expressed and purified a mutant form of Δ30-Msp1, Δ30-Msp1$^{E214Q}$, bearing a commonly used 'Walker B' mutation that inactivates ATP hydrolysis but leaves ATP binding intact. Δ30-Msp1 formed homogeneous hexamers as assessed by size-exclusion chromatography, whereas Δ30-Msp1$^{E214Q}$ formed hexamers but also higher order oligomers (*Figure 1—figure supplement 3*). We incubated Δ30-Msp1 with ADP•BeF$_x$, an ATP transition state analog that in ATPases (depending on whether BeF$_x$ is bound) can mimic both the ATP and the ADP states (*Monroe et al., 2017*), and Δ30-Msp1$^{E214Q}$ with ATP. Next, we prepared both samples for cryo-EM imaging and solved the structures of the hexameric assemblies.

3D classification of Δ30-Msp1 particles generated two distinct structures: In the first structure the Δ30-Msp1 hexamer adopted the right-handed open spiral arrangement characteristic of AAA proteins (20950 particles analyzed). Refinement of this Δ30-Msp1 hexamer structure yielded a map with an average resolution of 3.7 Å, approaching 3.0 Å in the stable core, with most side chain density in the complex well-resolved (*Figure 1—figure supplement 4*, *Table 1*). In the resulting model, six Δ30-Msp1 subunits (M1-M6) rotate and translate progressively to assemble into a right-handed open spiral, with an open seam between the top (M1) and the bottom (M6) subunits, similar to many other reported AAA protein structures (*Figure 1A*). In the second structure, the Δ30-Msp1 hexamer adopted the same spiral arrangement with the exception of M1, which showed less ordered density, indictive of a continuum of multiple unresolved states in the Msp1 reaction cycle (*Figure 1B*). We fitted M1 into its equilibrium position, in which it closes the seam of the spiral. This structure, henceforth referred to as the 'closed' conformation, was based on 48861 particles and resolved to 3.1 Å, approaching 2.5 Å in the stable core (*Figure 1—figure supplement 4*).

3D classification of Δ30-Msp1$^{E214Q}$ particles revealed two distinct structures (*Figure 1—figure supplement 5*). First, 45,687 particles contributed to a reconstruction of the hexamer in the open conformation at an average 3.5 Å resolution (approaching 3.0 Å at its core (*Figure 1—figure supplement 5B and C*). The structure closely resembles the open conformation of Δ30-Msp1

**Table 1.** Data collection, reconstruction, and model refinement statistics.

**Data collection**

| | Δ30-Msp1[E214Q] | Δ30-Msp1 | |
|---|---|---|---|
| Microscope | Titan Krios | Titan Krios | |
| Voltage (keV) | 300 | 300 | |
| Nominal magnification | 22500x | 22500x | |
| Exposure navigation | Stage shift | Stage shift | |
| Electron exposure ($e^-Å^{-2}$) | 70 | 70 | |
| Exposure rate ($e^-$/pixel/sec) | 7.85 | 7.85 | |
| Detector | K2 summit | K2 summit | |
| Pixel size (Å) | 1.059 | 1.059 | |
| Defocus range (μm) | 0.6–2.0 | 0.6–2.0 | |
| Micrographs | 1443 | 2502 | |
| Total extracted particles (no.) | 502534 | 902573 | |

**Reconstruction**

| | Δ30-Msp1[E214Q] | Δ30-Msp1 (closed) | Δ30-Msp1 (open) |
|---|---|---|---|
| EMDB ID | 20320 | 20318 | 20319 |
| Final particles (no.) | 45687 | 48861 | 29723 |
| Symmetry imposed | C1 | C1 | C1 |
| FSC average resolution at 0.143/0.5, unmasked (Å) | 4.6/8.2 | 4.1/7.8 | 6.8/9.6 |
| FSC average resolution at 0.143/0.5, masked (Å) | 3.5/4.0 | 3.1/3.6 | 3.7/4.1 |
| Applied B-factor (Å) | 89.9 | 83.7 | 70.8 |
| Final reconstruction package | cryoSPARC v0.55 private beta | | |
| Local resolution range | 2.8–6.0 | 2.5–5.5 | 2.5–6.0 |

**Refinement**

| | | | |
|---|---|---|---|
| PDB ID | 6PE0 | 6PDW | 6PDY |
| Protein residues | 1672 | 1469 | 1660 |
| Ligands | 10 | 11 | 13 |
| RMSD Bond lengths (Å) | 0.003 | 0.003 | 0.002 |
| RMSD Bond angles (°) | 0.685 | 0.671 | 0.639 |
| Ramachandran outliers (%) | 0.06 | 0.07 | 0.06 |
| Ramachandran allowed (%) | 12.25 | 10.63 | 10.90 |
| Ramachandran favored (%) | 88.69 | 89.30 | 89.04 |
| Poor rotamers (%) | 0.14 | 0.25 | 0.00 |
| CaBLAM outliers (%) | 6.09 | 6.74 | 6.86 |
| Molprobity score | 1.99 | 2.06 | 2.14 |
| Clash score (all atoms) | 7.40 | 9.27 | 11.29 |
| B-factors (protein) | 73.26 | 69.33 | 107.50 |
| B-factors (ligands) | 54.73 | 46.51 | 78.24 |
| EMRinger Score | 2.00 | 2.92 | 1.62 |
| Model refinement package | phenix.real_space_refine (1.13-2998-000) | | |

(RSMD = 0.867). Second, 38,165 particles contributed to a reconstruction of Msp1 showing extra density, indicative of the presence of one or more extra 30-Msp1$^{E214Q}$ subunits extending the spiral staircase (*Figure 1—figure supplements 5B* and *6*). Henceforth, we focus our analyses exclusively on the structure of the homogeneous hexamer, because we observed the higher oligomers only in the context of the E214Q mutation.

In each subunit the core AAA domain follows a linker domain (LD). In the intact protein, the LD would directly follow the transmembrane helix, which is not present in our Δ30 constructs (*Figure 1D,E*). The LD contains two helices, a long α-helix, we named 'α0', and a shorter helix α1, and two loops (L1 and L2) (*Figure 1D*). Helix α0 and L1 fold into a fishhook-shaped motif. In the open conformation, we observed significant density for α0 in subunits M1-M5, whereas α0 in M6 was disordered (*Figure 2A*). The α0 helices from M1-M5 are radially organized with their N-termini pointing to the spiral's center and display a positively charged surface toward the face where the Msp1 complex would approach the membrane (*Figure 2B and C*). Moreover, the N-terminal regions of the helices from M1-M5 converge in a central hub where they contact each other in what must be a staggered alignment forced by the pitch of the spiral, with M1 being closest and M5 farthest from

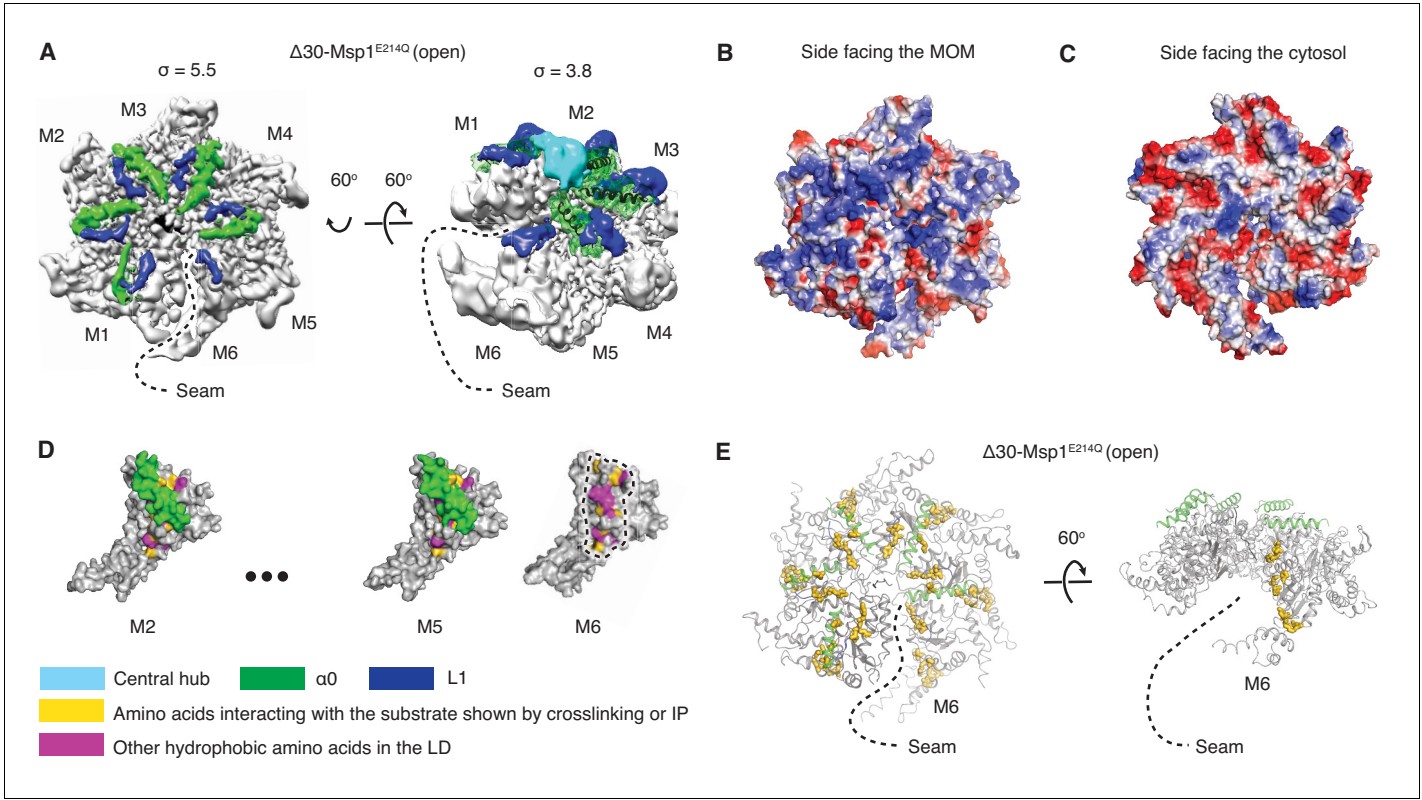

**Figure 2.** Structural details of the LD. (A) Cryo-EM map of Δ30-Msp1$^{E214Q}$ showing the arrangement of the fishhook motifs in the spiral. M1-M5 shows significant density for the entire fishhook motif (α0 and the L1), whereas M6 shows density for L1 but not for α0. On the left, the structure is displayed at σ = 5.5, showing the fishhook motifs of different subunits radially organized with their N-termini pointing to the center of the spiral. On the right, the structure is displayed at σ = 3.8, showing the density of the central hub (cyan) emerge where the α0s of M1-M5 converge in a staggered alignment. (B and C) The electrostatic potential surface of the Δ30-Msp1$^{E214Q}$ structure shows that Msp1 displays a positively charged surface. Positive charges are colored in blue, negative charges in red, and neutral side chains in white. (D) Surface representation of individual subunits highlighting amino acids in the LD likely to engage the hydrophobic substrate. These amino acids are buried in by α0 in M2-M5 but exposed in M6 where α0 is melted. The labeled amino acids include the previously identified L89, Y92, E93, V101, P103, I106, D112, I113, G114, G115, I116, and other hydrophobic amino acids L87, V88, V96, A97, L98, V100, A102, P107, V108, F110. (E) Mapping of amino acids that interact with the substrate (identified in *Li et al. (2019)* by crosslinking or immunoprecipitation) to the Δ30-Msp1$^{E214Q}$ structure shows that on M6, they form a patch at the seam of the spiral. The central hub is colored in cyan, α0 in green, L1 in blue, previously identified amino acids that interact with the substrate in gold, and other hydrophobic amino acids in the LD in magenta.

The online version of this article includes the following figure supplement(s) for figure 2:

**Figure supplement 1.** Peptide array and molecular modeling suggest Msp1's substrate specificity.

the membrane (*Figure 2A*, *Video 1*). We propose that for M6 this distance is too large for α0$_{M6}$ to participate in this interaction. This conjecture would explain why α0$_{M6}$ is melted, perhaps due to a lack of interactions in the central hub that stabilize α0$_{M1-5}$. Recent work identified a patch of hydrophobic amino acids on the LD to be required in substrate recruitment through co-immunoprecipitation and in vivo crosslinking. Msp1 bearing mutations in this region fail to pull down the substrate and also cause severe growth defects to cells lacking a functional GET pathway. The analyzed LD mutations did not affect hexamer formation, suggesting that the hydrophobic amino acids on the LD could constitute substrate recruitment sites in the Msp1 hexamer (*Li et al., 2019*). Mapping of the hydrophobic amino acids to our structure showed that most of them are shielded by α0s in M1-M5. (*Figure 2D*). By contrast, the hydrophobic patch becomes uniquely exposed due to the melting of α0$_{M6}$ in M6, that is on the subunit that caps the bottom end of the spiral next to the open seam (*Figure 2E*).

This notion suggests that Msp1 preferentially binds to hydrophobic peptides. To address this point directly, we tiled 10-mer peptides in an array, stepping through Msp1's known substrates and shifting three amino acids at a time. Plotting the fold-enrichment against amino acid hydrophobicity showed that hydrophobic amino acids were enriched (*Figure 2—figure supplement 1A and B*). Hydrophilic amino acids were disenriched, with the exception of arginine and lysine, perhaps due to forming cation-π interactions with aromatic amino acids in the hydrophobic patch in the LD (Y92, F110). We surmise that the binding assay does not discriminate between substrate interaction sites on Msp1 and that these results perhaps also reflect strong hydrophobic and cation-π interactions with the aromatic amino acids in the central pore discussed below.

## Pore-loops engage in an extensive web of substrate interactions in the central pore

Upon modeling the protein into the density, we observed in all three structures extra density in the central pore of the spiral, spanning its entire depth (*Figure 3A*). This density likely represents an averaged composite of a mixture of peptides from endogenous *E. coli* proteins that engaged with and became trapped in the central translocation pore. Modeling a linear 10-mer peptide into the density showed clear side chain features yet did not reveal side chain identities or the polarity of the putative translocation substrate. We modeled the putative peptide as poly-alanine with the C-terminus juxtaposed to M1 and its N-terminus to M6 (*Figure 2B*). The peptide adopts an extended conformation in which every two adjacent amino acids face in opposite direction resembling a β-strand and are then rotated around 60° to remain in register with the contacting subunits in the spiral.

In the open conformation, three pore-loops extend from each ATPase domain and shape Msp1's central pore (*Figure 3B and C*). Six pore-loops one form a spiral staircase around the substrate (*Figure 3D,E* and *Figure 3—figure supplement 1*, *Video 2*). The KX$_1$X$_2$G motif in pore-loop one is highly conserved across the entire AAA protein family, with X$_1$ being an aromatic and X$_2$ usually a non-aromatic hydrophobic amino acid. By contrast, in Msp1/ATAD1 both X$_1$ and X$_2$ are aromatic (Msp1: W187, Y188). Akin to other AAA proteins, W187 inserts between two side chains of the translocating peptide, with its ring orthogonal to the substrate peptide backbone. Y188 interacts with two different substrate side chains, which lie on opposing sides of the substrate chain. Its aromatic ring lies parallel to the peptide backbone (*Figure 3E*). The main chain NH of Y188 forms a hydrogen bond with the substrate's backbone carbonyl (*Figure 3D*), further fortifying the interaction. Mutating W187 or Y188 to alanine causes significant growth defect in yeast, confirming their importance (*Figure 4*).

By contrast to pore-loops one which are well-ordered and contribute to substrate contacts from all six subunits, pore-loops two in M1 and M6 are disordered, reflective of the special conformational status of the subunits that cap the spiral ends (*Figure 3C and H*). In addition, while ordered, pore-loop two on M5 is disengaged from the substrate (*Figure 3D*). Pore-loops 2

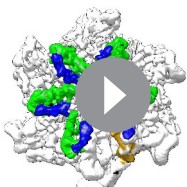

**Video 1.** The linker domain (LD) of Msp1.
https://elifesciences.org/articles/54031#video1

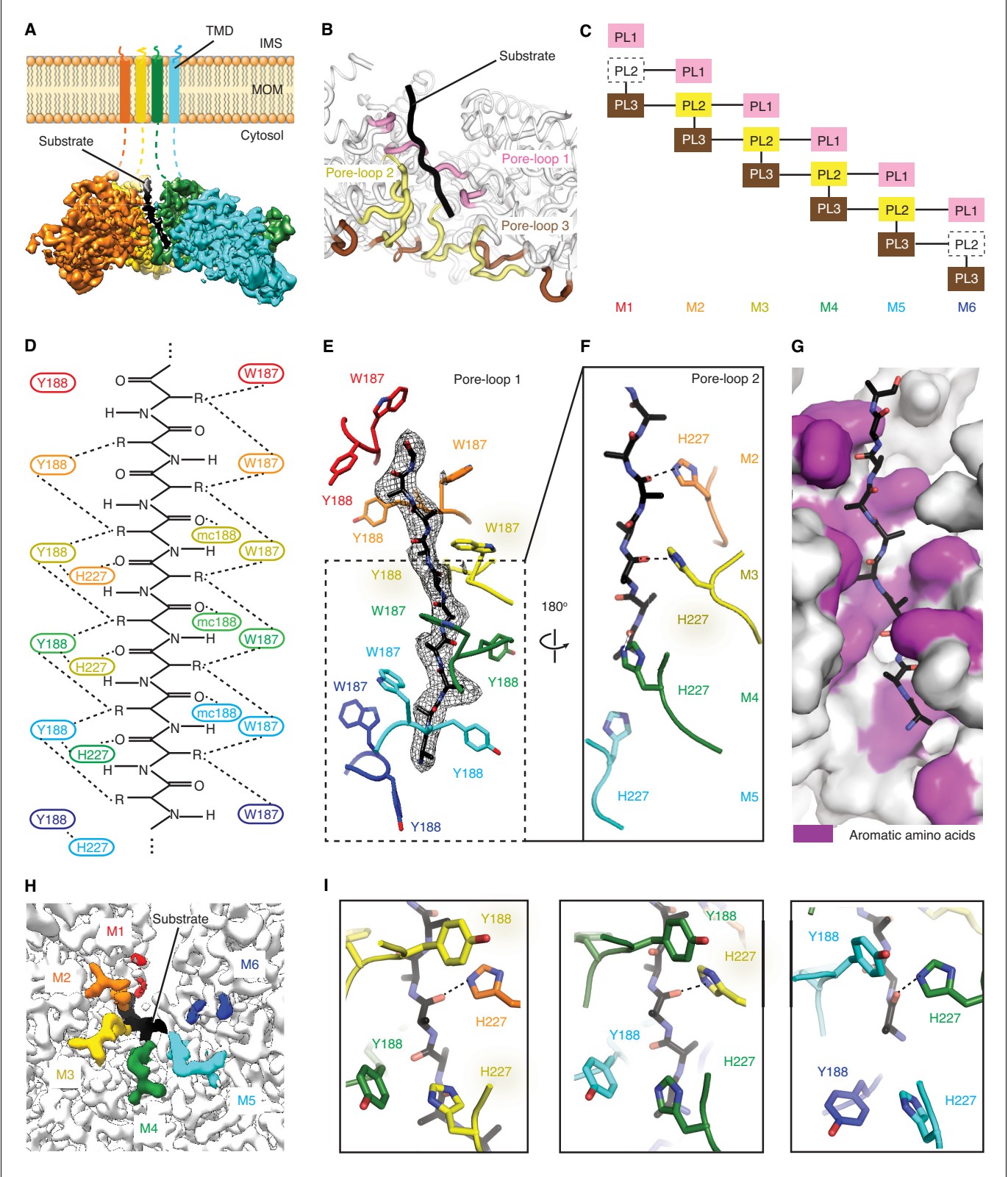

**Figure 3.** Msp1 interacts with the substrate via unique pore-loops. (A) Cut-away view of the Δ30-Msp1^E214Q map showing the substrate density (highlighted in white dashed lines) in the central pore. (B) Cartoon representation of the three pore-loops. Pore-loop one is shown in pink, pore-loop two in yellow and pore-loop three in brown. (C) Schematic diagram showing the interactions between pore-loops. Each line represents one interaction. The disordered pore-loops two in M1 and M6 are shown in boxes with dashed lines. (D) Schematic diagram showing that the pore-loops interact with

*Figure 3 continued*

the substrate through both side chain and main chain (mc) contacts. (E) Pore-loops one form a staircase around the substrate. The peptide density is shown in black mesh. (F) Pore-loops two form a second staircase below pore-loops 1. H227s form hydrogen bonds with the peptide backbone carbonyls (dashed lines). (G) Surface representation of the central pore, showing that the peptide (in stick representation) is surrounded by aromatic amino acids (colored in magenta) in the central pore. (H) Cryo-EM map showing the view of the central pore. Pore-loops 2 of M2-M5 are well-ordered, and those in M1 or M6 disordered. Pore-loops two are colored the same as in *Figure 1*. The substrate peptide is colored black. (I) Zoomed-in views of the peptide binding pockets showing that the substrate's side chain is inserted into a tetrameric-aromatic cage formed by two pairs of interlocking Y188-H227 sidechains. From left to right are the tetrameric-aromatic cages formed by M2-M4, M3-M5, and M4-M6.

The online version of this article includes the following figure supplement(s) for figure 3:

**Figure supplement 1.** Structural details of the pore-loops' interactions with the substrate.

**Figure supplement 2.** Pore-loop two forms a web of interactions with pore-loops 1 and 3 from subunits on both sides.

from M2-M5 form a second staircase below pore-loops 1 (*Figure 3F* and *Figure 3—figure supplement 1*). In M2-M5 pore-loops 2, H227 (conserved in Msp1 and ATAD1) hydrogen-bonds to the backbone carbonyl of the substrate, one peptide bond removed from the carbonyl that interacts with main chain NH of Y188 described above (*Figure 3D and F*). In addition, H227 forms π−π stacking bonds to Y188 (from pore-loop 1) of the clockwise-adjacent subunit, stabilizing the parallel alignment of the Y188 aromatic ring to the substrate backbone (*Figure 3D and I*, *Video 2*). We validated the functional importance of pore-loop 2 by mutating several amino acids (including R222, E226 and E228) in this loop. The mutants severely diminished Msp1 activity (*Figure 4*). By contrast, cells expressing Msp1 with mutations at the H227 position show no growth defect, perhaps because the two aromatic amino acids (W187 and Y188) in pore-loop 1 account for the majority of Msp1's grip on the substrate.

Pore-loops 3 are more distant from the pore center and do not contact the substrate directly. Rather, two pore-loops 3 from adjacent subunits encase each pore-loop 2 in an electrostatic network. In this arrangement, R222 on pore-loop 2 is sandwiched between pore-loops three from the same and the counter-clockwise positioned subunits (*Figure 3—figure supplement 2*). Because pore-loop 2 H227 also stacks with pore-loop 1 of the clockwise subunit, pore-loop 2 centrally contributes to a web of interactions that intimately link the subunits (*Figure 3C*, *Figure 3—figure supplement 2*). This arrangement explains why pore-loops 2 in M1 and M6, that is the subunits next to the spiral's seam are disordered due to their lack of stabilization on either side (*Figure 3C*). In line with our observation, mutations to the amino acids involved in this electrostatic network (including R222, R264 and D269) inactivate Msp1 (*Figure 4*), confirming their functional importance.

Upon close examination of the central pore, we noticed that regardless of position, every substrate side chain is always surrounded by aromatic pore-loop amino acids (*Figure 3G*). Substrate side chain on one side of the central pore are sandwiched by two tryptophans (W187; *Figure 3D and E*) and the ones on the other side face a tetrameric aromatic cage formed by two pairs of interlocked tyrosine/histidine pairs (Y188-H227; *Figure 3I*, *Video 2*). Indeed, when we modeled the top hits from the peptide array into the central pore both cationic and hydrophobic amino acids fitted comfortably between the staircase of W188 side chains and inside the tetrameric aromatic cages (*Figure 2—figure supplement 1C to E*).

## Subunits along the spiral propagate a linear sequence of nucleotide states in the reaction cycle

In Δ30-Msp1$^{E214Q}$, M1-M5 display clear density for ATP, whereas M6 does not contain significant density for a nucleotide, again indicating that M6 is in a distinct conformation from the other

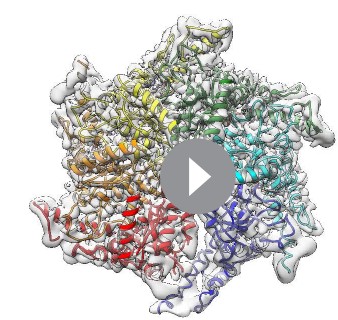

**Video 2.** The substrate interactions in the central pore.
https://elifesciences.org/articles/54031#video2

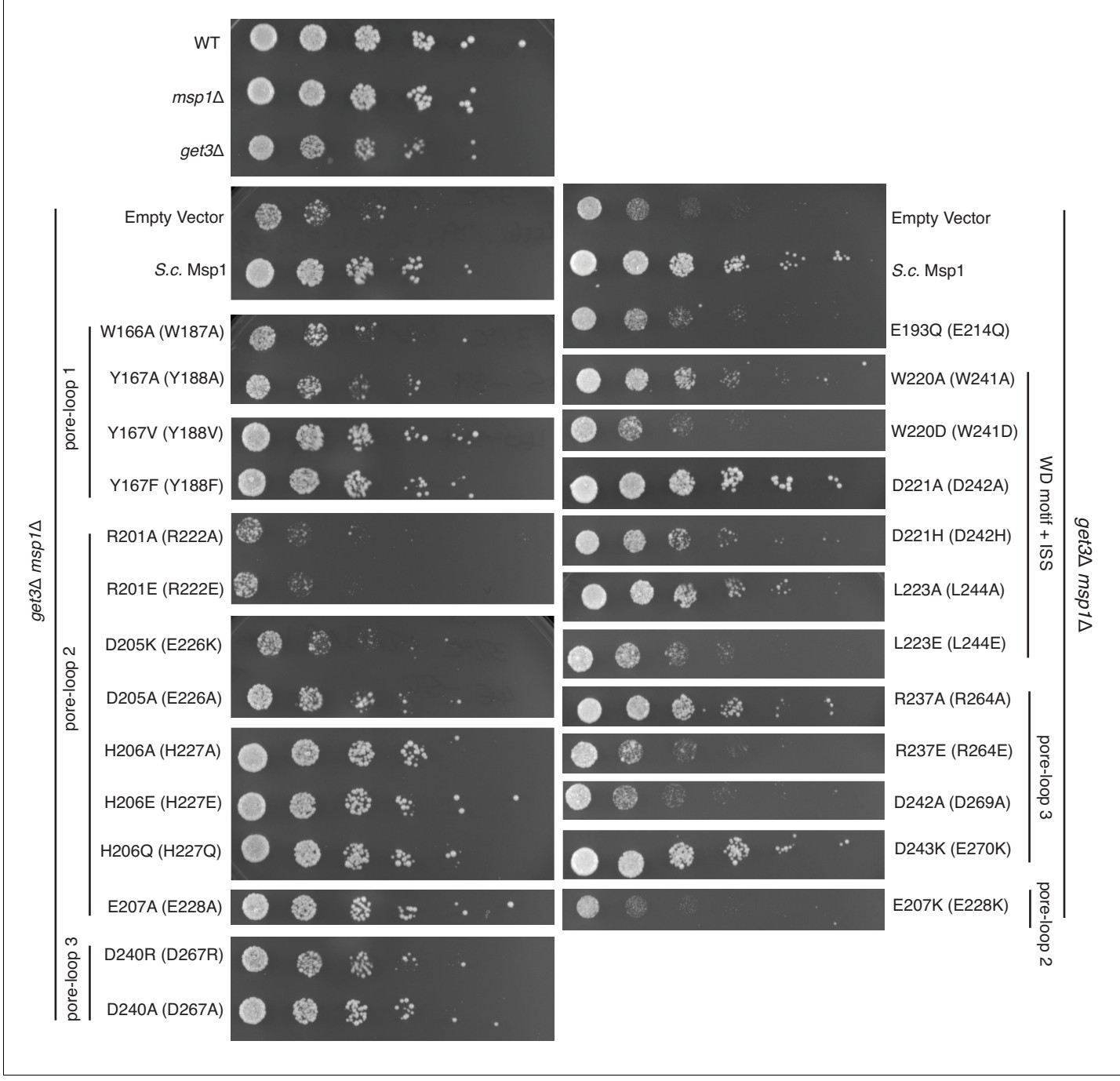

**Figure 4.** Yeast growth assays. Yeast growth assay showing mutations in the pore-loops, the WD motif and the ISS disrupt Msp1's activity in vivo. All mutations are introduced to the *S. cerevisiae* Msp1 (*S.c.* Msp1) in the *get3Δ msp1Δ* background. The corresponding amino acid numbers in *C. thermophilum* are shown in parentheses. All the strains are grown on SD-URA plates at 37° C. This image is a representative of N = 3 trials.

subunits (*Figure 1C*, *Figure 5—figure supplement 1*). The well-resolved nucleotide-binding pockets in the Δ30-Msp1•ADP•BeF$_x$ structures allowed us to unambiguously assign the nucleotide-bound states for most subunits (*Figure 5—figure supplement 1*). In the open conformation, M1 -M4 were bound to ADP•BeF$_x$ (mimicking the ATP state), M5 was bound to ADP, and M6 contained an empty nucleotide-binding pocket. In the closed conformation, each subunit displayed the same nucleotide states, except for M1, for which we could not assign a nucleotide state due to its multiple

conformations that resulted in a mostly disordered map (*Figure 1B*). These analyses point at an ordered sequence of ATP binding, hydrolysis, and release in the sequential subunits M1-M6 along the spiral.

The sequence of nucleotide states reveals mechanistic insight into how ATP hydrolysis is linked to the subunits' movements along the spiral track. Yme1 and the proteasomal AAA proteins have a conserved phenylalanine in the ISS that forms strong π- π stacking with three other phenylalanines in the opposing subunit. This contact is crucial in the tight coupling of the ATP hydrolysis to the pore-loop retraction. By contrast, in $AAA_{MC}$ proteins the conserved phenylalanine in the ISS motif is replaced by an aliphatic amino acid (L244 in Msp1; *Figure 1—figure supplement 1*) that cannot undergo π- π stacking interactions. Instead, Msp1 L244 forms hydrophobic interactions with two phenylalanines (F175 and F211; the third phenylalanine is replaced by an asparagine, N177, in Msp1) in the opposing subunit, weakening their interaction. This suggests that in Msp1 and other $AAA_{MC}$ proteins, the ISS may not be the main structural element that transmits the nucleotide state change. Rather, we observed that an adjacent tryptophan-aspartate pair (WD motif) interacts closely with two arginines (R274 and R275; called 'arginine fingers' in AAA proteins) that form cation-π and ionic interactions between W241 and R274, and D242 and R275, respectively (*Figure 5A–C*). R274 and R275 insert from the clockwise-apposed subunit into the ATP binding pocket of the adjacent subunit and stably bond to the triphosphate of ATP. Upon loss of the γ-phosphate by ATP hydrolysis in M5, we observed that the arginine fingers in M6 become less ordered, as does the interacting M6 WD motif (*Figure 5*, A, E and F). This leads to the melting of a structured loop unique to Msp1, lifting it from the surface of M5 (*Figure 5*, E and F). We named this loop, which may be unique to Msp1, the <u>n</u>ucleotide <u>c</u>ommunication <u>l</u>oop (NCL). At the ATP-bound intersubunit interface, the NCL stacks on top of L2 (amino acids 100–109, the short linker that follows α1 and is a structural feature unique to the $AAA_{MC}$ proteins, *Figure 1D*) of the counter-clockwise adjacent subunit. The interactions between the two structural elements are mainly hydrophobic. Thus, the stacking between the back-bone of the NCL and the hydrophobic side chains (such as P107, I106 and V101) is predicted to exclude water molecules and to achieve a gain in entropy, both favoring this conformation.

Melting of the NCL reduces the buried surface area between M6 and M5 to 74 $Å^2$, compared to 98 $Å^2$ between adjacent M1-M5 subunits. The Δ30-Msp1$^{E214Q}$ structure serves as a convenient control: M5 in Δ30-Msp1$^{E214Q}$ binds to an ATP instead of ADP and the corresponding WD motif and NCL in M6 are both well-ordered (*Figure 5D*), supporting our hypothesis that the NCL melts in response to ATP hydrolysis. Thus, Msp1 ATP hydrolysis and phosphate release triggers conformational changes relayed by the arginine fingers and the WD motif that result in melting of the NCL and weakening of the M5-M6 intersubunit interaction.

The fact that the NCL undergoes a nucleotide state-dependent conformational change would not necessarily rule out the importance of the degenerate ISS. To test this notion, we made mutations in L244 of the ISS. The L244A mutation causes a mild growth defect in yeast, and the L244E mutation causes a significant growth defect, suggesting that the hydrophobic interaction between the degenerate ISS and the counter-clockwise adjacent subunit also contributes to the Msp1's function.

Our mechanistic model can explain a recently identified disease-related mutation. The conserved aspartate (D221) in the WD motif in the human ATAD1 is found mutated to histidine in some schizophrenia patients. ATAD1$^{D221H}$ exhibits an oligomer disassembly defect, which reduces its ability to regulate AMPA receptor trafficking in neurons and causes impaired memory and social behavior in mice (*Umanah et al., 2017*). Based on our structures, replacing D242 (in *C. thermo*, D221 in humans) with an aromatic amino acid may increase its affinity toward the arginine fingers due to strong cation-π interactions. Also, $π - π$ stacking between W241 and the histidine would likely lead to a higher intrinsic stability of the mutated motif, and thus a more stable NCL. The well-folded NCL would result in a stronger interaction between subunits, impeding the movement of the spiral along its substrate(s), causing the observed defect. Mutating the WD motif results in a growth defect in yeast, confirming their functional importance (*Figure 4*).

Whether the mitochondrial localized ATAD1 acts on the plasma membrane localized AMPA receptors by residing at a junction site of the two membranes, or by redistributing to a different subcellular localization in neurons remains unclear.

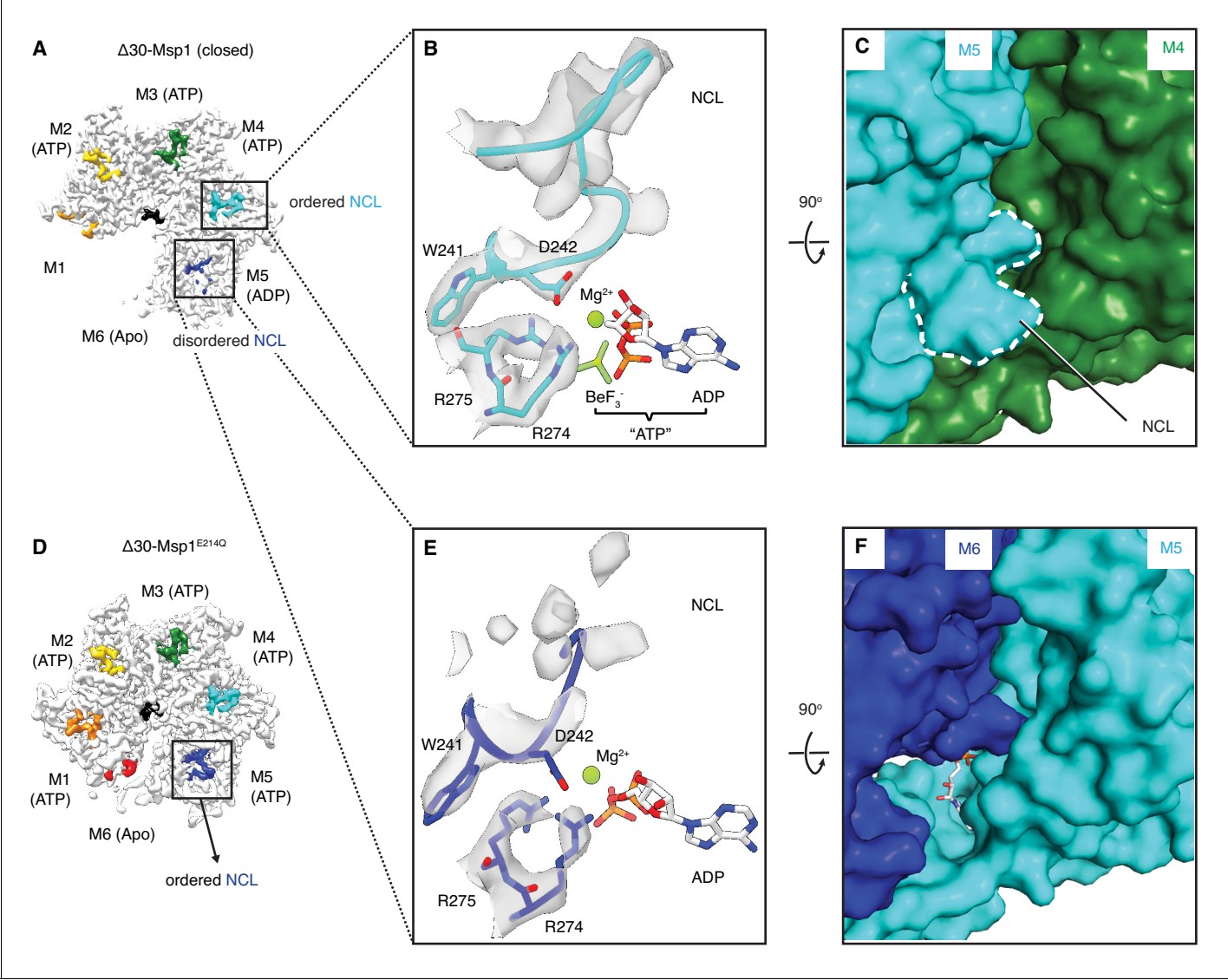

**Figure 5.** The NCL communicates the nucleotide-bound state between adjacent subunits. (A) The cryo-EM map of Δ30-Msp1 showing that the NCLs interacting with the ATP-bound subunits (M2–M4) are well ordered, whereas those interacting with the ADP (M5) or the Apo (M1) subunits are disordered. (B) Map of the nucleotide-sensing elements in M5 showing well-ordered arginine fingers (R274, R275), WD motif (W241 and D242) and the NCL. (C) 90° rotated view of (B) showing the surface representation of the M4-M5 interface. The NCL in M5 is highlighted in dashed lines. (E) Map of the nucleotide-sensing elements in M6, showing its less rigid arginine fingers, WD motif, and disordered NCL. (F) 90° rotated views of (E) showing the surface representation of the M5-M6 interface. (D) The cryo-EM map of Δ30-Msp1$^{E214}$ showing that the NCL of M6 is well-ordered as it senses the ATP-bound state of M5. The Msp1 subunits and their corresponding NCLs are colored as in *Figure 1*. Nucleotides and BeF$_3^-$ are shown in stick representation and colored by element. Mg$^{2+}$ is shown as a green spheres.

The online version of this article includes the following figure supplement(s) for figure 5:

**Figure supplement 1.** Structural details of nucleotide binding pockets in the Δ30-Msp1 (closed) complex.

## Discussion

We resolved three structures of the AAA protein Msp1 at average resolutions between 3.1 Å – 3.7 Å (*Figure 1*, A to C). Based on these high-resolution structures that capture Msp1 in the act of translocating a substrate polypeptide through its central pore, we propose a model of the mechanism by which Msp1 acts in a series of coordinated events (*Figure 6*). Our data inform on the three stages of AAA protein/substrate interactions: 1) the initial substrate recruitment; 2) the intersubunit communication coupled with ATP hydrolysis, and 3) the stepwise M1-M6 subunit/substrate interactions along

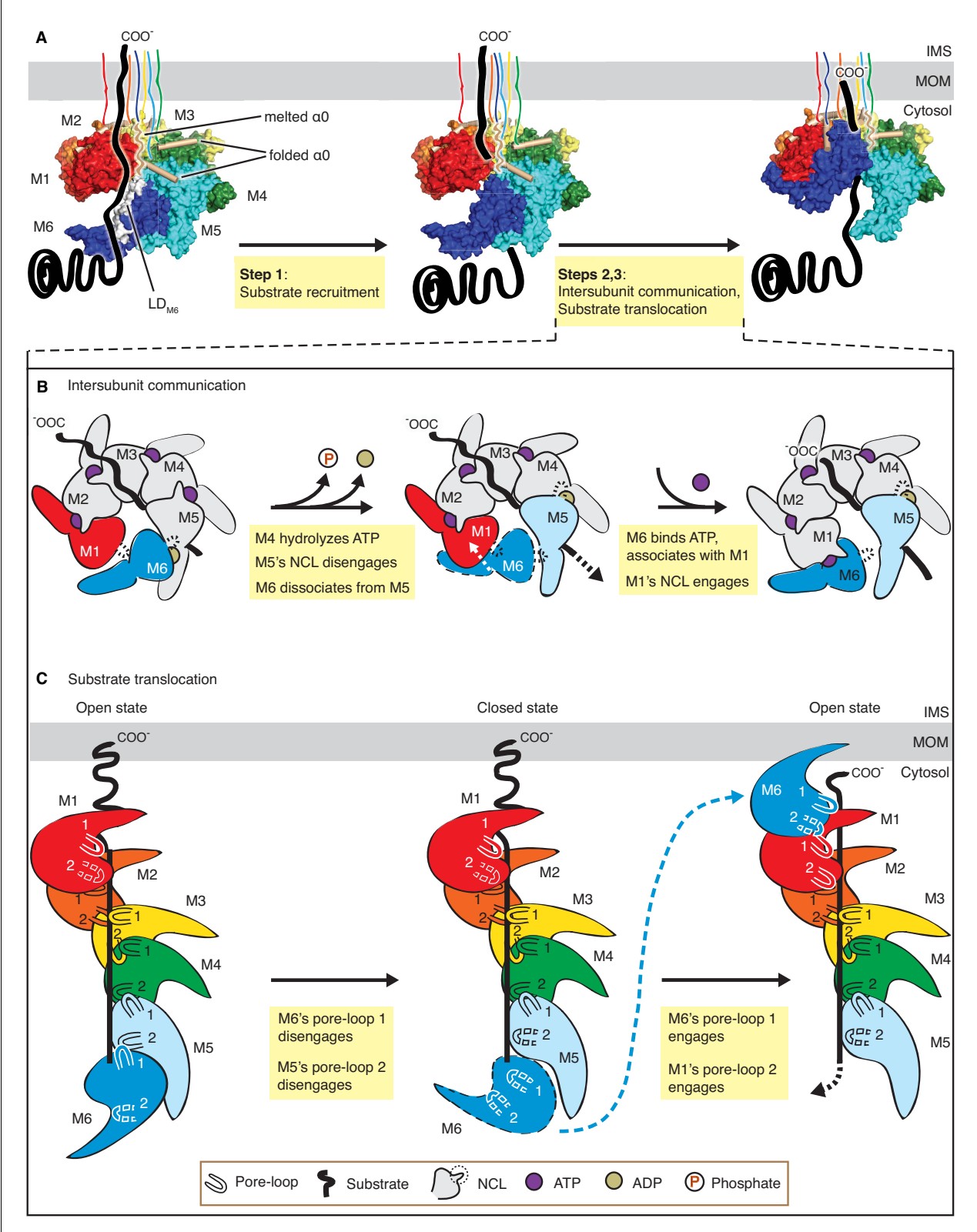

**Figure 6.** Mechanistic model for Msp1-mediated peptide extraction. (**A**) Model for Msp1's mechanism illustrated in three major steps. The Msp1 models on the left and the middle are of Δ30-Msp1$^{E214}$; the right one is generated by rotating the Δ30-Msp1$^{E214Q}$ model counter clockwise by one monomer. A model TA protein substrate is shown in black, with its C-terminal tail inserted in the membrane (shown as a gray bar). The folded α0 is shown in cylinder representation, and the melted one in squiggly lines. Msp1 subunits are colored the same as in *Figure 1* and α0 in tan. The positions

*Figure 6 continued on next page*

*Figure 6 continued*

of Msp1's N-terminal transmembrane regions are schematically indicated. (**B**) Schematic model for the NCL-mediated inter-subunit communication. The disordered subunit and the dislodged NCLs are outlined with dashed lines. (**C**) Schematic model for substrate translocation through the central pore, showing the sequential disengagement of pore-loops 1 and 2 at the bottom (M6) position and the sequential engagement at the subunit at the top (M1) position. The disordered subunit and pore-loops are outlined in dashed lines.

The online version of this article includes the following figure supplement(s) for figure 6:

**Figure supplement 1.** Comparison of ISS and NCL.
**Figure supplement 2.** Cryo-EM map of the ISS motif in M1.
**Figure supplement 3.** Overlay of the Δ30-Msp1 (closed) to the Vps4-substrate complex structures.

the substrate's translocation path (*Figure 6A*). The emerging mechanistic details at each stage expand our general understanding of AAA protein functions and elucidate features that specialize AAA$_{MC}$ subfamily members, and Msp1 in particular. We will discuss these three stages in turn.

## Substrate recruitment

The substrate-binding site previously mapped to the LD has remained structurally ill-defined for most AAA proteins. For the AAA$_{MC}$ subfamily member katanin, the LD contains a characteristic 'fish-hook' module, previously resolved at 3.5 Å (*Zehr et al., 2020*). Our higher resolution structure (3.1 Å; *Figure 1B*) allowed us to build a high-confidence molecular model. The model resolves two conformational states, one state is observed in M1-M5, in which α0 is well folded and shields the subunits' hydrophobic substrate binding sites identified through co-immunoprecipitation and in-cell cross-linking (*Li et al., 2019*). The other state is only observed in M6, in which α0 is disordered and the substrate-binding site is exposed (*Figure 2A and D*). Intriguingly, the LD$_{M6}$ surface-exposed substrate binding site is juxtaposed to the open seam (*Figure 2E*). Our data suggests a plausible mechanism where such juxtaposition serves to align bound substrate with the seam from where it can be conveniently threaded into the pore (*Figure 6A*, Step 1). In this way, a singular hydrophobic binding site on the Msp1 hexamer could select a substrate with exposed hydrophobic properties, which would be expected for a mistargeted membrane protein lacking appropriate interaction partners. According to this notion, the subunit occupying the M6 position would always have the hydrophobic binding site exposed and hence potentially interact with more substrates even when the central pore is occupied. A plausible mechanism would then assume that, when the central pore is unoccupied, this interaction aligns the substrate with the seam mediating its entry, whereas, when the central pore is occupied, the additional substrate would not be threaded in. Rather, in the pore-occupied state the upward translocation of M6 refolds α0, and α0's interactions with the patch's hydrophobic residues compete off the additional substrate. In summary, our structures combined with the biochemical identification of the hydrophobic binding site on the LD (*Li et al., 2019*) suggest a possible mechanism of Msp1's substrate recruitment. While it seems reasonable to assume that entry would occur through the open seam, the details of substrate entry and threading into the central pore remains to be determined.

## Intersubunit communication

Based on our model (*Figure 6B*), a linear sequence of nucleotide states propagates along the Msp1 subunits in the open spiral. The NCL, identified here as a short insertion positioned C-terminally juxtaposed to the traditionally defined ISS motif (*Figure 1—figure supplement 1*), is ideally positioned to communicate the nucleotide-bound state and perhaps direct ATP hydrolysis according to the subunit's position in the spiral. Such communication occurs in two stages: first (Stage 1), at the M5-M6 interface, the arginine fingers and the WD motif initiate a series of allosteric changes by detecting the loss of the γ-phosphate group in ATP. Similar changes are also observed in other AAA proteins, such as Yme1 (*Puchades et al., 2017*) and Vps4 (*Han et al., 2017*); yet by contrast, the consequences of these conformational changes in Msp1 diverge from those in other AAA proteins at the second stage of the process (Stage 2): In Yme1, for example, M6 (numbered according to the Msp1 structure, see *Figure 6—figure supplement 1*) is the only ADP-bound subunit, and the counter-clockwise adjacent subunit M1 is mobile. In Stage 2 at Yme1's M1-M6 interface, M1's arginine finger initiated allosteric changes in response to ATP hydrolysis in M6 lead to M1 retracting a conserved

phenylalanine from its stacking partners in M6 (*Figure 6—figure supplement 1D,G,J*). The loss of this strong π-π stacking interaction causes M1 to depart from M6 (*Figure 6—figure supplement 1J*). In Stage 2 in Vps4, there are two ADP-bound subunits (M5 and M6, *Figure 6—figure supplement 1B*). At the M1-M6 interface, the ISS motif undergoes similar retraction from its neighbor, indicating its role in communicating the nucleotide-bound state as described in Yme1 (*Figure 6—figure supplement 1K*). However, at the M5-M6 interface, although bound with ADP, the position of the ISS is almost identical to that at the ATP-bound one (M4-M5, *Figure 6—figure supplement 1E and H*), with a valine (which replaces the phenylalanine in Yme1) deeply inserted into its counter-clockwise subunit. Therefore in Vps4, the ISS motif does not respond to ATP hydrolysis at the M5-M6 interface.

By contrast, we observed only a single ADP-bound subunit (M5, *Figure 1C*, *Figure 6—figure supplement 1C*). At the M5-M6 interface, as described in detail in *Figure 5*, the arginine fingers sense ATP hydrolysis and results in the loss of rigidity in the WD motif, which leads to the melting of the NCL. As observed for the M5-M6 interface in Vps4, the position of the ISS remains unchanged from the ATP-bound interface, with a leucine (valine in Vps4, and phenylalanine in Yme1) stably inserted into M6 (*Figure 6—figure supplement 1F and I*). Rather, it is the NCL (i.e. the short loop insertion that follows the ISS) that undergoes significant conformational changes (*Figure 6—figure supplement 1I*, *Figure 5*) that weaken the M6-M5 interaction and prepare the terminal M6 for its departure from the spiral assembly. In other words, our structure captured a novel state not present in other reported AAA protein structures, in which the terminal subunit M6 becomes predisposed for translocation by the melting of the NCL. Sequence alignment shows that the length of the NCL varies across Msp1 homologs, as well as among other AAA$_{MC}$ proteins (*Figure 1—figure supplements 1* and *2*). The *C. thermophilum* Msp1 used in this study has a long NCL, as well as the mammalian ATAD1, katanin and spastin, whereas the *S. cerevisiae* Msp1 and Vps4 both have short ones. As described above, the short loop that follows the ISS in Vps4 is not able to respond to nucleotide-state change at the M5-M6 interface the same way as in the *C. thermophilum* Msp1, and the short loop that follows the ISS is also too short to contact its neighbor (*Figure 6—figure supplement 1H*). Therefore, it is likely that the nucleotide state-dependent conformational change observed in our structures require an NCL long enough to touch the counter-clockwise adjacent subunit, and our structures could serve as a model for the mammalian ATAD1 and similar AAA$_{MC}$ proteins like katanin. We noticed that the W in the WD motif is strictly conserved in Msp1/ATAD1 (*Figure 1—figure supplement 2*), but variable in the AAA$_{MC}$ family, suggesting that Msp1 might have evolved this additional sensing mechanism to enhance the coupling between the nucleotide state and the movement of the subunit.

The WD motif and NCL-mediated intersubunit communication is a unique feature that differs from what has been observed for AAA proteins. The recently published structure of Yme1 suggested a mechanism by which the ISS-mediated refolding of an α−helix and subsequent retraction of the pore-loops allosterically transmits information regarding the nucleotide state to the central pore. An alternative mechanism has been proposed for spastin (*Sandate et al., 2019*) where the ISS does not seem to engage in the intersubunit communication but, rather, an electrostatic network may connect the nucleotide-binding pocket to the central pore. To assess whether Msp1 could use a combination of these mechanisms in addition to the NCL, we compared the conformation of Msp1's ISS at multiple intersubunit interfaces. We did not observe the refolding of the α−helix that proceeds the ISS at the M5-M6 (ADP- bound) interface. The quality of the map at the M1-M6 (apo) interface is not sufficient to conclude whether a helix refolding event took place (*Figure 6—figure supplement 2*). Likewise, we also did not observe a significant conformational change of pore-loop 3 (which is the main component of the electrostatic network that connects the nucleotide binding pocket and the substrate-interacting pore-loops) (*Figure 3—figure supplement 2*), indicating that it is insensitive to the change in the nucleotide states. Rather, as mentioned above, the structures supports the hypothesis that pore-loops 3 help form an interconnected network to enable pore-loops 2 to sense the position of the subunit (*Figure 3C*) and help initiate the upward translocation of M6.

The degenerate ISS motifs (where an aliphatic amino acid replaces the phenylalanine in DGF) are not only present in the AAA$_{MC}$ proteins (*Figure 1—figure supplement 1*) but also in more distally related proteins, such as NSF (DGV) and paraplegin (DGM, *Figure 1—figure supplement 1*). Accordingly, high-resolution structures of these proteins (including spastin [*Sandate et al., 2019*], katanin (*Zehr et al., 2020*), Vps4 (*Han et al., 2017*), NSF (*White et al., 2018*) and this work) have

converged on the observation that their ISS motifs do not undergo ATP hydrolysis-dependent conformational change and that alternative mechanisms must account for such communication. The NCL-mediated intersubunit communication described here may be a unique feature to Msp1/ATAD1. Thus, there may not be a universal mechanism that applies to all proteins with a degenerate ISS.

## Substrate translocation

The structure of the substrate in the translocation pore revealed an extensive interaction network and tight pore dimensions. When removing the substrate computationally, we measured the diameter of Msp1's central pore at ~8 Å, indicating that it is significantly narrower than pores in other substrate-bound AAA protein structures (e.g. ~13 Å for Vps4 [*Han et al., 2017*], *Figure 6—figure supplement 3*). Compared to other AAA proteins, a tryptophan from pore-loop 1 and an interlocked tyrosine-histidine (Y188-H227) pair contributed by pore-loops 2 from two adjacent subunits create a hydrophobic environment in the central pore with ample potential for hydrophobic and π-stacking interactions (*Figure 3*, E, F and I). It is important to note, that the proposed hand-over-hand mechanism by which the spiral translocates on the substrate does not require subunits in the M2-M5 positions to loosen their grip on the substrate (*Figure 5C*). The tight network of substrate/pore-loop interactions in M2-M5, including the sequence-promiscuous hydrogen bonding interactions between the pore-loop 2 H227 and the substrate's backbone (*Figure 3D and F*), thus do not need to be broken, allowing Msp1 to exert a high degree of processivity that intuitively would avoid substrate backsliding. The recently published structures of Yme1 (*Puchades et al., 2017*), Rix7 (*Lo et al., 2019*) and AFG3L2 (*Puchades et al., 2019*) showed that pore-loops 2 form an additional staircase around the substrate below pore-loops 1 as observed in Msp1, suggesting a potential role in substrate griping; yet, neither structure showed a direct contact (polar or hydrophobic) between pore-loop 2 and the substrate. By contrast to the intimate substrate interactions of pore-loops 1 in both structures, Y396 of pore-loop 2 in Yme1 points away from the substrate and is on average ~5 Å away from the substrate. The same is true for Rix7, where pore-loop 2 encases the substrate, but the distance in between the loop and the substrate (~4–6 Å) is too large for a direct interaction. By contrast, several structures of the AAA$_{MC}$ proteins spastin (*Sandate et al., 2019*), katanin (*Zehr et al., 2020*) and Msp1 (this work) all showed a direct interaction between amino acids in pore-loops 2 and the substrate. The three structures together converge on the functional importance of pore-loops 2 of AAA$_{MC}$ proteins, which is different from other AAA proteins in which pore-loops 2 do not directly engage the substrate.

With three aromatic amino acids from each subunit, Msp1's pore-loops are particularly bulky. A recent study on the ClpXP motor showed that the bulkiness of the pore-loops is positively correlated to the grip on the substrate but inversely correlated to the substrate pulling velocity (*Rodriguez-Aliaga et al., 2016*). It is perhaps beneficial for Msp1 to exert more force on its substrate, which contain hydrophobic membrane anchors that require a larger force to extract from the lipid bilayer. Similar to Msp1, Cdc48 (*Cooney et al., 2019*; *Twomey et al., 2019*), a AAA protein that extracts ubiquitinated proteins from the ER membrane also uses a double-aromatic pore-loop1 to intercalate the substrate. Also, the mitochondrial inner membrane AAA proteases Yme1 (*Puchades et al., 2017*) and AFG3L2 (*Puchades et al., 2019*) both have an aromatic amino acid in pore-loop 2 (in addition to the conserved aromatic amino acid in pore-loop 1) that intercalates the substrate, which suggests that distally related AAA proteins may have converged on similar solutions to increase their grip on extracting membrane protein substrates.

In addition to the substrate interactions, we also observed that pore-loop 2 is involved in a web of interactions engaging pore-loops 1 and 3 across adjacent subunits (*Figure 3C*, *Figure 3—figure supplement 2*). As evident in all three Msp1 structures, pore-loop 2 is well-ordered only when its stacking partners exist on both sides (*Figure 3C and H*). This property allows it to detect the subunit's position in the spiral: pore-loop 2 becomes disordered and breaks its interaction with the substrate when it is in the M6 position by detecting the absence of the stacking partner on the opposite side of the seam (*Figure 3C*). Having loosened its grip on the substrate and also dislodged its NCL (*Figure 6B and C*), M6 dissociates from the spiral complex. It then samples multiple states between the M1 top and the M6 bottom positions of the spiral, until it is loaded with a new ATP molecule. In this way, the outgoing M6 subunit repositions itself in the M1 position on the opposite side of the seam (*Figure 6A and C*). This rebuilding process extends the spiral on the top end while shrinking it

at the bottom end. This upward translocation pushes the membrane and the remaining five subunits in opposite directions, thereby extracting two amino acids of the substrate from the membrane. As these amino acids enter the pore, they interact with pore-loop 1 of the new subunit in the M1 position (*Figure 6A and C*). By contrast, pore-loop 2 on this subunit remains disordered and only engages the substrate once the next subunit cycles into the M1 position. As an M6 subunit dissociates to moves to the M1 position, its previously disordered α0 refolds into a helix (*Figure 6A*), accommodating the vertical movement of M6, while allowing its TMD to remain stably inserted in the membrane.

In conclusion, Msp1's structural elements illuminate the process by which the membrane-bound enzyme utilizes a functionally adapted core engine to recognize and extract its protein membrane-bound substrates. The overall mechanism is in strong agreement with many recently published AAA protein structures, including the spiral arrangement of the subunits, the sequential ATP hydrolysis around the ring and the hand-over-hand substrate translocation. In additional to the conservation in the overall mechanism, the structures suggest that Msp1 utilizes an elegant mechanism of lateral substrate alignment at the opening seam, and evolved a particularly strong coupling between ATP hydrolysis and substrate movement through its central pore. Many of the mechanistic details revealed here for Msp1 are likely applicable to other AAA$_{MC}$ proteins and pave the way to understand specialization in AAA proteins in general.

# Materials and methods

## Key resources table

| Reagent type (species) or resource | Designation | Source or reference | Identifiers | Additional information |
|---|---|---|---|---|
| Gene (*Chaetomium thermophilum*) | *Msp1* | Uniprot | G0S654 | |
| Genetic reagents (*S. cerevisiae*) | *MATα leu2-3,112 TRP1 can1-100 ura3-1 ADE2 his3-11,15* (wild-type) | PMID: 24821790 | PWY1944 in the lab stock | |
| Genetic reagents (*S. cerevisiae*) | *msp1Δ::HpH$^R$* | PMID: 24821790 | PWY1947 in the lab stock | |
| Genetic reagents (*S. cerevisiae*) | *get3Δ::NAT$^R$* | PMID: 24821790 | PWY1950 in the lab stock | |
| Genetic reagents (*S. cerevisiae*) | *msp1Δ::HpH$^R$ get3Δ::NAT$^R$* | PMID: 24821790 | PWY1953 in the lab stock | |
| Recombinant DNA reagent | GST-thrombin-*C.thermo* Msp1 (plasmid) | This paper | | Materials and method section: cloning of Msp1 |
| Recombinant DNA reagent | GST-thrombin-*C. thermo* Msp1 (E214) (plasmid) | This paper | | Materials and method section: cloning of Msp1 |
| Software, algorithm | MotionCor2 | PMID: 28250466 | RRID: SCR_016499 | |
| Software, algorithm | Relion | PMID: 23000701 | RRID: SCR_016274 | |
| Software, algorithm | Cryosparc | PMID: 28165473 | RRID: SCR_016501 | |
| Software, algorithm | UCSF Chimera | PMID: 15264254 | RRID: SCR_004097 | |
| Software, algorithm | GCTF | PMID: 26592709 | RRID: SCR_016500 | |
| Software, algorithm | Phenix | PMID: 20124702 | RRID: SCR_014224 | |
| Software, algorithm | Coot | PMID: 20383002 | RRID: SCR_014222 | |
| Software, algorithm | Pymol | Schrödinger, LLC | RRID: SCR_000305 | |

## Cloning of Msp1

To generate the construct used for cryo-EM studies, the gene encoding the cytosolic domain of *C. thermophilum* Δ30-Msp1 was PCR amplified and subcloned into a pGEX-2T vector encoding an N-terminal GST tag followed by a thrombin cleavage site. To generate the construct for peptide array, the same Msp1 gene was PCR amplified and subcloned into a pET28 vector encoding an N-terminal 6xHis tag followed by a thrombin cleavage site. The Walker B mutation (E214Q) is

introduced by site directed mutagenesis. All the constructs are verified by DNA sequencing. To generate the constructs used in the yeast growth assays, the *MSP1* ORF flanked with its upstream 1000 bp promoter region was PCR amplified from yeast genomic DNA and ligated into the pRS416 vector. Mutations were introduced by site-directed mutagenesis of the wild-type Msp1 construct.

## Protein purification

The plasmid encoding the GST-tagged Msp1 is transformed into *E. coli* BL21 (DE3). Cells were grown at 37°C overnight in LB media overnight, before diluted into 1-l culture. Protein expression was induced by adding 0.5 mM IPTG when $OD_{600}$ reached around 1.0. Cells were harvested after 4 hr of expression at 37°C. The cell pellets were resuspended in Msp1 lysis buffer (25 mM HEPES pH 7.5, 300 mM NaCl, 1 mM DTT, 2 mM $MgCl_2$) supplemented with the protease inhibitor cocktail (Roche) and lysed by Emulsiflex. The crude lysate was clarified by centrifugation at 30,000 x g for 30 min at 4°C. The supernatant was then incubated with glutathione beads (Pierce) for 3 hr at 4°C. The glutathione beads were washed with 15 column volumes (CV) of Msp1 lysis buffer, and eluted with 5 CV of lysis buffer supplemented with 20 mM glutathione.

Thrombin protease (GE Healthcare) was added to the elution and incubated at 4°C overnight to allow complete proteolytic removal of the GST tag. The resulting mixture was loaded onto a size exclusion column (SEC) (Superdex 75 16/600, GE Healthcare) in the Msp1 lysis buffer. Fractions corresponding to Msp1 were pooled and concentrated before loaded to the second SEC column (Superdex 200 10/300, GE Healthcare). Fractions corresponding to Msp1 were again pooled and concentrated to around 200 µM, flash frozen in liquid nitrogen and stored at −80°C before further use. Both the Δ30-Msp1 and the Δ30-Msp1$^{E214Q}$ proteins were purified as described above.

His-Δ30-Msp1$^{E214Q}$ was expressed in the same way as GST-Δ30-Msp1. The cell pellets were resuspended in the lysis buffer (25 mM HEPES pH 7.5, 300 mM NaCl, 10 mM βME, 20 mM imidazole, 2 mM $MgCl_2$). The procedures for lysing the cells and clarifying cell lysates were the same as those for GST-Δ30-Msp1$^{E214Q}$. The supernatant was loaded onto the Ni-NTA resin (Qiagen) that was washed with the lysis buffer. The mixture was incubated in a gravity column for 30 min at 4°C. The Ni-NTA resin was washed with 10 CV of lysis buffer, and the protein was eluted with 7 CV of elution buffer (lysis buffer supplemented with additional imidazole to make a final concentration 300 mM). The resulting mixture was loaded onto a SEC column (Superdex 200 10/300, GE healthcare). Fractions corresponding to Msp1 were again pooled and concentrated to around 200 µM, flash frozen in liquid nitrogen and stored at −80°C before further use.

## Sample preparation of cryo electron microscopy

Msp1 was diluted to 50–100 µM into buffer containing 25 mM HEPES pH 7.5, 300 mM NaCl, 1 mM DTT, 2.5% glycerol and the appropriate nucleotide (2 mM ATP and 2 mM $MgCl_2$ for Δ30-Msp1$^{E214Q}$ or 5 mM $MgCl_2$ and 5 mM ADP•BeF$_x$ for Δ30-Msp1. BeSO$_4$ and KF was mixed at 1:5 molar ratio to generate BeF$_x$, which is then mixed with ADP in equal molar ratio to generate ADP•BeF$_x$). The sample was incubated on ice for 1–2 hr before plunge freezing. A 3 µl aliquot of the sample were applied onto the Quantifoil R 1.2/1/3 400 mesh Gold grid and incubated for 15 s. A 0.5 µl aliquot of 0.1–0.2% Nonidet P-40 substitutes was added immediately before blotting using the Whatman #1 blotting paper. The entire blotting procedure was performed using Vitrobot Mark IV (FEI) at 10°C and 100% humidity. The grids were not glow discharged.

## Electron microscopy data collection

Cryo-EM data was collected on a Titan Krios transmission electron microscope operating at 300 keV and micrographs were acquired using a Gatan K2 summit direct electron detector. The total electron exposure was 70 e⁻/ Å$^2$, fractioned over 100 frames during a 10 s exposure. Data was collected at 22,500 x nominal magnification (1.059 Å/pixel at the specimen level) and nominal defocus range of −0.6 to −2.0 µm.

## Image processing

For Δ30-Msp1$^{E214Q}$, the micrograph frames were aligned using MotionCorr2. The contrast transfer function (CTF) parameters were estimated with GCTF (*Zhang, 2016*). Particles were automatically picked using Gautomatch and extracted in RELION (*Scheres, 2012*) using a 256-pixel box size.

Images were down-sampled to a pixel size of 4.236 Å and classified in 2D in RELION. Classes that showed clear protein features were selected and extracted with re-centering and then imported into cryoSPARC (*Punjani et al., 2017*) for *ab initio* reconstruction (k = 3). Homogeneous refinement was performed on the best model to yield a reconstruction of 8.92 Å. This structure was used together with the three structures from *ab initio* reconstruction for heterogeneous refinement. The structure resulting from the best class was refined and used for a new round of ab initio reconstruction. The best model from the new *ab initio* reconstruction was subjected to homogeneous and then heterogeneous refinement (as described above) for three more rounds to yield two major species: the hexamer and the larger oligomer. Homogeneous refinement of the two structures yielded structures of 3.9 Å/3.8 Å, respectively. These two structures together with a low-resolution model resulting from the first round of heterogeneous refinement were used as input models for a final round of heterogeneous refinement against a particle stack corresponding to the best class of the first round of heterogeneous refinement. The resulting hexamer/larger oligomer structures were refined with homogeneous refinement ++ (as implemented in cryoSPARC v0.5.5-privatebeta), yielding the final reconstructions of 3.5 Å/3.7 Å, respectively.

For Δ30-Msp1, every step through 2D classification was performed in the same way as Δ30-Msp1$^{E214Q}$. After three rounds of 2D classification and selecting the good classes, the selected particles were extracted with re-centering and subjected to 3D classification in RELION, using the Δ30-Msp1$^{E214Q}$-hexamer structure as a reference. Refinement of the best class generated a consensus structure of 3.7 Å, where M2-M6 were well resolved and M1 has poor density. An additional round of 3D classification on this structure was performed without alignment to identify the two major species: Δ30-Msp1 (open) and Δ30-Msp1 (closed). Particles corresponding to each species were extracted with re-centering and imported to cryoSPARC for refinement. Homogeneous refinement ++ was used to generate the final reconstructions of the two structures at 3.7 Å/3.1 Å resolution, respectively.

## Atomic model building and refinement

The big and the small AAA domain of the crystal structure of the monomeric *S. cerevisiae* Msp1 (PDB ID: 5W0T) was used to generate the predicted structures of the *C. thermophilum* Msp1 in SWISS-MODEL (*Schwede et al., 2003*). The six big AAA domains and the six small AAA domains were individually docked into the map of Δ30-Msp1$^{E214Q}$ in *Chimera* (*Pettersen et al., 2004*) using the *Fit in Map* function. The resulting model was subjected to rigid body refinement in Phenix (*Adams et al., 2010*), again treating each AAA domain as an individual rigid body. M4 was used for initial real space refinement in *Coot* (*Emsley et al., 2010*) because it showed the best resolution among all monomers. Missing linkers and loops were built de novo using *Coot* and Phenix. The resulting model of M4 was rigid body fitted into the density of the other five monomers, and residue-by-residue refinement was performed in *Coot*. The final models of subunits were combined into the full hexamer with the ATP molecules modeled into the nucleotide binding pockets. Additional density was observed in the central pore corresponding to the trapped peptide. Although showing clear side chain features, we could not assign the side chain identities, therefore we modeled a poly-alanine sequence into the density, with the C-terminus facing the membrane side of the protein. For Δ30-Msp1 (closed), after building models for M2-M6, M4's big and small AAA domains were fitted into the density of M1 individually. ADP was first modeled into nucleotide binding pockets of M2-M5. M2-M4 showed significant extra density in which we modeled the BeF$_3^-$ ion. The real space refinement was performed in a similar way to Δ30-Msp1 (closed). The model of Δ30-Msp1 (open) was built in a similar way to Δ30-Msp1 (closed). The figures displaying structures are prepared with PyMOL (*Schrodinger LLC, 2015*) and *Chimera*.

## Peptide array

The peptide array was purchased from the MIT Biopolymers Laboratory. The peptide is attached to the membrane through a PEG$_{500}$ linker via an amide linkage. The array was composed of 10-mer peptides that were tiled along Msp1's known substrates with a three amino acid shift between adjacent spots. The Msp1 substrates are: the human Pex26 and Gos28, the *S. cerevisiae* Pex15 and Gos1, the *C. thermophilum* Gos1, and the cytosolic portion of the human GluR2 (GluR2C). The array was washed with methanol for 10 min and then with protein buffer (25 mM HEPES pH 7.5, 300 mM

NaCl, 2 mM MgCl$_2$, 1 mM TCEP, 0.02% Tween 20) 3 times for 10 min each. An aliquot of His-$\Delta$30-Msp1$^{E214Q}$ was diluted into 10 ml of protein buffer and the diluted protein was incubated under room temperature for 5 min before applied to the array. After 2 min of incubation with the array, ATP was added to the final concentration of 2 mM. The array was incubated at room temperature for another hour. Then the array was washed three times with the wash buffer (protein buffer supplemented with 2 mM ATP), each for 10 min to remove the unbound protein. Using a semi-dry apparatus, bound Msp1 was electrophorectically transferred to a nitrocellulose membrane and detected with anti-His$_6$ (Abcam) antibody. The binding intensity in each spot was normalized to the strongest signal intensity in the peptide array. The peptides with the top 20% binding scores were pooled together to calculate the occurrence of each amino acid. The values are normalized to their abundance in the input.

## Molecular modeling

$\Delta$30-Msp1 (closed) was chosen to perform molecular modeling, because it has the highest resolution among the three structures presented in this work. Peptide binding conformations were calculated using the PyRosetta package (*Chaudhury et al., 2010*). Peptide sequences were threaded onto the poly-alanine backbone in the $\Delta$30-Msp1 (closed) structure. Since the exact binding positions were unknown, each sequence was threaded with nine different shifts from −4 to 4. The overhanging parts were removed and missing parts were appended with alanines. For example, when the sequence DHWKSFRNIR was threaded with shift −2, the peptide sequence used in simulation was WKSFRNIRAA. For each threaded peptide, 200 trajectories of Rosetta fast relax simulation (*Conway et al., 2014*) were performed using the ref2015 score function (*Alford et al., 2017*). The fast relax method repacked side chains and minimized the structure in a simulated annealing. Each trajectory comprised of three fast relax repeats. During the simulation, the peptide, pore-loops 1 and 2 (amino acids 185–189 and 222–228) of each subunits are movable while the rest part of the complex was kept fixed. Extra rotamers were enabled by the -ex1 -ex2 flags. The lowest energy conformations were recorded for further analysis.

## Yeast growth assay

The wild-type, *get3$\Delta$*, *msp1$\Delta$* and *get3$\Delta$ msp1$\Delta$ S. cerevisiae* strains were obtained as described previously (*Okreglak and Walter, 2014*). The standard lithium acetate procedure was used for yeast transformation. Transformed yeast cells were grown in synthetic complete dextrose (SD) medium lacking uracil at 30℃. Transformed cells were grown overnight in SD-Ura media. Cultures were diluted to OD$_{600}$ is about 0.1 and grown at 30℃ for 3 hr. The resulting culture was again diluted to the same OD$_{600}$, serially diluted 5X in SD-Ura, and spotted onto SD-Ura plates, and grown at 37℃.

## Acknowledgements

We thank A Frost, V Belyy and Z Chen for critical reading of the manuscript; V Okreglak, D Southworth for helpful discussions; M Braunfeld, D Bulkley of the UCSF Center for Advanced CryoEM facility, which is supported by NIH grants S10OD021741 and S10OD020054 and the Howard Hughes Medical Institute (HHMI); Z Yu, R Huang and H Chou of the CryoEM Facility at the HHMI Janelia Research Campus; the QB3 shared cluster for computational support.

## Additional information

### Funding

| Funder | Grant reference number | Author |
| --- | --- | --- |
| National Institutes of Health | R01GM032384 | Lan Wang Peter Walter |
| Howard Hughes Medical Institute | | Peter Walter |
| Damon Runyon Cancer Research Foundation | DRG-2312-17 | Lan Wang |

The funders had no role in study design, data collection and interpretation, or the decision to submit the work for publication.

## Author contributions
Lan Wang, Conceptualization, Data curation, Formal analysis, Investigation, Visualization, Methodology, Writing - original draft, Writing - review and editing; Alexander Myasnikov, Data curation, Investigation, Methodology; Xingjie Pan, Data curation, Investigation; Peter Walter, Conceptualization, Supervision, Funding acquisition, Writing - original draft, Writing - review and editing

## Author ORCIDs
Lan Wang  https://orcid.org/0000-0002-8931-7201
Peter Walter  https://orcid.org/0000-0002-6849-708X

## Decision letter and Author response
Decision letter https://doi.org/10.7554/eLife.54031.sa1
Author response https://doi.org/10.7554/eLife.54031.sa2

# Additional files

## Supplementary files
• Transparent reporting form

## Data availability
All data needed to evaluate the conclusions in the paper are present in the paper and/or the supplementary materials. The atomic models were deposited in the protein data bank under the accession codes 6PDW, 6PDY and 6PE0; the associated cryo-EM maps were deposited in the electron microscopy data bank (EMDB) under the accession codes EMD-20318, EMD-20319 and EMD-20320.

The following datasets were generated:

| Author(s) | Year | Dataset title | Dataset URL | Database and Identifier |
|---|---|---|---|---|
| Wang L, Myasnykov A, Pan X, Walter P | 2020 | Msp1 (E214Q)-substrate complex | http://www.rcsb.org/structure/6PE0 | RCSB Protein Data Bank, 6PE0 |
| Wang L, Myasnykov A, Pan X, Walter P | 2020 | Msp1 (E214Q)-substrate complex | http://www.ebi.ac.uk/pdbe/entry/emdb/EMD-20320 | Electron Microscopy Data Bank, EMD-20320 |
| Wang L, Myasnykov A, Pan X, Walter P | 2020 | Msp1-substrate complex in closed conformation | http://www.rcsb.org/structure/6PDW | RCSB Protein Data Bank, 6PDW |
| Wang L, Myasnykov A, Pan X, Walter P | 2020 | Msp1-substrate complex in closed conformation | http://www.ebi.ac.uk/pdbe/entry/emdb/EMD-20318 | Electron Microscopy Data Bank, EMD-20318 |
| Wang L, Myasnykov A, Pan X, Walter P | 2020 | Msp1-substrate complex in open conformation | http://www.rcsb.org/structure/6PDY | RCSB Protein Data Bank, 6PDY |
| Wang L, Myasnykov A, Pan X, Walter P | 2020 | Msp1-substrate complex in open conformation | http://www.ebi.ac.uk/pdbe/entry/emdb/EMD-20319 | Electron Microscopy Data Bank, EMD-20319 |

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
