## [Decision Letter]

**Acceptance summary:**

This study presents cryo-EM structures of the AAA protein Msp1 in different nucleotide states and bound to substrate peptides. Many AAA proteins act as ATP-hydrolyzing protein remodelers by threading substrate proteins through the central pore of their hexameric assemblies. Msp1 extracts mis-localized proteins from the mitochondrial outer membrane and targets them for degradation. Msp1 belongs to the so-called meiotic clade of the AAA protein family. It is unique in comprising an N-terminal trans-membrane helix. The protein is thought to recognize mis-localized membrane proteins by hydrophobic sequences exposed at the membrane. The authors present cryo-EM structures of Msp1 from Chaetomium thermophilum at 3.1 Å-3.7 Å resolution. Both pore loops 1 and 2, and indirectly pore loop 3, contribute to interactions with the bound substrate, suggesting an intriguing mechanism for the function of Msp1 in extracting mis-localized membrane proteins.

**Decision letter after peer review:**

[Editors’ note: the authors submitted for reconsideration following the decision after peer review. What follows is the decision letter after the first round of review.]

Thank you for submitting your work entitled "Structure of the AAA protein Msp1 reveals mechanism of mislocalized membrane protein extraction" for consideration by *eLife*. Your article has been reviewed by three peer reviewers, and the evaluation has been overseen by a Reviewing Editor and a Senior Editor. The following individuals involved in review of your submission have agreed to reveal their identity: Peter Chien (Reviewer #1); Gabriel C Lander (Reviewer #2).

Our decision has been reached after consultation between the reviewers. Based on these discussions and the individual reviews below, we regret to inform you that the current version of the manuscript will not be considered further at this time for publication in *eLife*. As you will see below, this decision is based on the time it is anticipated that requested experiments would take to complete rather than on the suitability of the study for *eLife*.

All reviewers and the reviewing editor agreed that your study, once thoroughly revised and amended, should be a strong candidate for publication in *eLife*. As you can see, the reviewers concur that the study lacks mutational follow-up experiments to validate key conclusions of the structural analysis by identifying residues in Msp1 that are critical for activity. We prefer (and encourage) re-submission rather than revision in order to give you sufficient time to perform such an analysis and maintain the opportunity to go through another round of revisions. Mutant analysis in yeast should specifically address the suggested role of the NCL and also the function of pore loop 2 (see reviewer #1 and point 6 of reviewer #2). We believe that such additional data would substantially improve the paper.

Reviewer #1:

In this work, the authors reveal high resolution cryo-EM structures of the Msp1 oligomer complex in several states. Msp1 is an outer mitochondria membrane localized protein that extracts mistargeted tail-anchored proteins when they are not properly inserted into the ER. While loss of Msp1 in yeast shows strong fitness effects only when combined with loss of the GET pathway, loss of the mammalian ATAD1 ortholog has profound effects on neurons. Therefore, it is of high interest to understand how this quality control pathway mechanistically operates.

Here, the authors determine structures of Msp1 bound to substrate in closed and open states, proposing a cycle of ATP hydrolysis and coordinated pore loop engagement that extracts membrane targeted substrates. Mutations confirm the general principles that grip, and ATP hydrolysis are important for function in yeast and detailed exploration of the structure suggests some very interesting features (such as an exposed hydrophobic seam region) that make Msp1 selective for its function. However, beyond the higher resolution structures, there was little testing of the hypotheses proposed here – even the mutagenesis work was limited to those residues already shown in past work to be important (Wohlever et al., 2017), with the inclusion of a single new mutation suggested by this current structure (Arg201). It also seems one of the major conclusions of the work is that the ordered-disordered transition of the NCL is an important part of the work cycle, but this was not directly tested experimentally. This lack of follow up characterization reduced my overall enthusiasm for the work.

Although there are a number of AAA family structures that have been solved by cryo-EM recently, this work stands out because Msp1 has a highly specialized function requiring a number of unique features. Like other members of the Meiotic-Clade, there is a wide seam in the Msp1 ring upon substrate binding that is seen in both katanin (Zehr et al., 2017) and spastin (Sandate et al., 2019). The authors propose that this hydrophobic seam is important for engaging mistargeted substrates that are improperly targeted to the mitochondria outer membrane. This is a very intriguing hypothesis and is consistent with previous work from this lab and others, but there is no testing of this hypothesis showing the role of this hydrophobic seam in engagement or extracting substrates. Ideally, some combination of mutagenesis in Msp1 or substrate would allow for experimental validation of this model for the hydrophobic seam.

The claim that the pore-loop 2 contacts with substrates are a completely new feature of this work is interesting. A number of studies have shown a role for pore loop 2 (or equivalent pore loops) in stabilizing engaged substrate, however this work reveals intimate direct contacts between sidechains in Msp1 with the backbone of putative substrates (specifically H227 as shown in Figures 3 D/F). This is intriguing in that these direct interactions have not been seen in other AAA structures – in fact, some AAA machines do not seem to care about the specific nature of the polypeptide backbones. For example, ClpX can translocate polymers with up to 10 methylene groups separating consecutive peptide bonds (Barkow et al., 2009) suggesting the specific placement of the carbonyls is not critical for substrate engagement. The current structures suggest more intimate contacts between Msp1 and its substrate that seem like they would constrain substrates more strictly. Ideally, determining whether this is actually the case experimentally would strengthen the impact of this almost exclusively structural study.

Reviewer #2:

Wang et al. present and discuss cryo-EM structures of Type I AAA+ protein Msp1 bound to peptide substrate. As observed in numerous other substrate-bound structures of AAA+ protein remodelers, Msp1 forms a spiral staircase around the peptide substrate. Structural mobility of an Msp1 construct lacking the N-terminal membrane-associated domains was minimized for structure determination using two different approaches: 1) introducing a Walker B mutation or 2) incubating Msp1 with non-hydrolyzable ATP analog. The authors present two distinct conformational states that are similar to those previously determined for katanin, another AAA+ ATPase from the meiotic clade. However, the resolution of these reconstructions were substantially higher than those determined for katanin, which enabled the authors to examine the molecular details of the substrate processing mechanism. These data indicate a conservation of the hand-over-hand model for substrate translocation in Msp1, offering snapshots of the dynamic process by which the bottom-most subunit of the AAA+ staircase transitions to the top-most position as part of the ATP hydrolysis cycle. The authors identify unique features of the N-terminal domain of the enzyme that might be involved in recruitment of substrate at the membrane interface and demonstrate a preference for hydrophobic substrate peptides. The authors also propose that Msp1 communicates nucleotide state from one subunit to the next through a mechanism that is distinct from those proposed for other AAA+ proteins, involving sequences that flank the previously identified ISS motif. While the results are noteworthy and intriguing, the proposed mechanism of allostery requires biochemical validation, as well as further analysis and discussion in the context of the AAA+ superfamily.

Major points:

1) The authors propose a mechanism of allostery that largely centers on the melting of a post-ISS loop they refer to as the NCL. The mechanistic details of this loop's melting and its relevance to intersubunit interactions are unclear. Can the authors define the interactions that the NCL that stabilize intersubunit interactions? Further, the quality of the cryo-EM density should enable the authors to specify the residue-specific responses to the repositioning of the arginine fingers and the WD motif that results in its melting and should be described. Importantly, the functional relevance of these residues should be verified using the established yeast growth assay (or other biochemical assays). Given that the NCL is highly variable in sequence and length among meiotic clade AAA+ proteins, the case for an NCL-based mechanism that is conserved across this subfamily is not particularly strong.

2) The authors identify a WD motif that is very likely to be involved in the mechanism of action of this and other meiotic clade AAA+ proteins. However, the D from this motif is also a conserved component of the ISS, and it was previously proposed in YME1 that the D of the ISS is involved in sensing the position of the consecutive Arginines. Given that the W residue is not conserved across the meiotic AAA+ clade, the functional relevance of the W in this motif should be further explored through mutagenesis. How the corresponding residue might be relevant to function in other meiotic clade ATPases should also be speculated.

3) As shown by the authors in Figure 1—figure supplements 1 and 2, in meiotic clade AAA+ proteins, pore loops 2 and 3 contain conserved positive and negatively charged residues that have been shown to be required for activity. The authors further describe a powerful interconnected network formed by positive and negatively charged residues in pore loops 2 and 3 (Figure 3—figure supplement 3), which is also seen in our structure of spastin (Sandate et al., 2019). In spastin, this network connects the pore loop of one subunit to the nucleotide-binding pocket of its counterclockwise neighbor, suggesting that this charge network might be the main driver of intersubunit communication and allosteric transmission of nucleotide state to the pore loops, which explains their essential role for activity. Could this also be the case in Msp1? Can the authors include the nucleotides in their current Figure 3—figure supplement 3? The authors should also evaluate the functional relevance of this pore loop charged network in Msp1 using their yeast growth assay (i.e. alanine substitutions of the conserved charged residues in the pore loops).

4) Are changes in the nucleotide binding pocket being transmitted to the pore loops, or do the authors propose that the pore loops release from substrate as a result of the rigid-body displacement of the subunit as it transitions from the bottom to the top of the spiral?

5) As discussed by the authors, the ISS motif (DGF) has emerged as an allosteric driver of inter-subunit communication in numerous AAA+ proteins. The authors claim that the substitution of DGF for DGL diminishes the role that the ISS motif plays in intersubunit coordination in Msp1, due to loss of π-stacking interactions with the core β-strands of the adjacent subunit. However, both F and L are hydrophobic residues, and the loop in Msp1 indeed appears to extend into the hydrophobic groove at the intersubunit interface where the L could engage in hydrophobic interactions (Figure 5—figure supplement 1). I agree that the absence of π-stacking interactions likely decreases the relevance of the ISS motif itself in the mechanism of allostery of Msp1, but the potential relevance of this loop to the Msp1 mechanism cannot be discounted. Further, ADP-BeF is known to mimic a transition state in AAA+ proteins, and it is thus reasonable to believe that the disordered, but unretracted ISS motif observed in the ADP-like subunit in fact corresponds to a transition state wherein the ISS motif is in a transitional refolding state. To gain a better understanding of this loop's functional role, the authors should incorporate the following:

a) Evaluate the functional relevance of DGL in Msp1 using their yeast growth assay (or other biochemical assays) to determine the effect of DGL◊DGA (and ideally DGE and DGV mutations).

b) Expand their description of the nucleotide binding pocket of Msp1. Is the aromatic residue on the core β-strand that π-stacks with the ISS motif F in YME1/AFG3L2/Ftsh/26S proteasome not present in Msp1 and/or other meiotic clade AAA+ proteins? What interactions is DGL involved in, in addition to the D interaction with the consecutive arginines?

c) Include a figure showing the EM density for the ISS motif in the apo subunit and try to assess whether it undergoes a refolding event. I understand that the quality of the reconstruction for this mobile subunit is not sufficient for modeling, but the formation of a secondary element might be visible even at resolutions close to 8 Å.

6) I agree with the authors' overarching conclusion that Msp1 has evolved to function as a more powerful unfoldase through: i) double aromatic in pore loop 1, ii) substrate interacting pore loop 2, iii) a unique mechanism of allostery. This is the most important finding of this study, and I encourage the authors to expand their conclusion to include similar observations across the AAA+ superfamily:

a) The degenerate ISS motif and the consequences for the mechanism of allostery are a very important part of the mechanism of Msp1. In addition to Type I AAA+ proteins of the meiotic clade, degenerate ISS motifs are also present in other classical AAA+ proteins, such as Type II AAA+ protein NSF (DGV instead of DGF, White et al., 2018) and AAA+ protease paraplegin (DGM instead of DGF, Figure 1—figure supplement 1). Can the authors discuss their findings in the context of the role of degenerate ISS motifs across the AAA+ superfamily?

b) Increased bulkiness of the pore loops was shown to affect both the chemical and mechanical properties of the AAA+ motor (Rodriguez-Aliaga et al., 2016). Can the authors discuss their findings regarding the pore loops in this context?

c) The presence of two aromatic residues in pore loop 1 is noteworthy, and the authors should reference the recent papers of Type II AAA+ protein Cdc48 bound to substrate (Cooney et al., 2019, Twomey et al., 2019) that showed this same organization, and discuss how distantly related AAA+ proteins have converged on similar solutions to increase their grip on substrate.

d) Similarly, pore loop 2 has recently been shown to directly contact the substrate in the mitochondrial inner membrane AAA+ protease AFG3L2, which also appears to have evolved to be a more powerful unfoldase (Puchades et al., 2019). An expanded discussion on pore loop 2 interactions in different meiotic clade AAA+ proteins, as well as distantly related AAA+ proteases, to those in Msp1 would increase the impact of the authors' findings.

7) In Figure 1—figure supplement 5, the authors discuss the presence of heptamers in their sample, which might be an artifact of particles being aligned with an offset of 1 subunit register relative to one another. Have the authors attempted to further classify this heptameric class into subsets? In our recent work with the closely related Type I AAA+ protein spastin (Sandate et al., 2019), we also initially observed a heptameric reconstruction, but found that this organization was a processing artifact (described in the Materials and methods). Given that both spastin and Msp1 are single-ring Type I ATPases of the meiotic clade, and the spastin "heptamer" we observed contained a Walker B mutation, the authors may be encountering the same register misalignment.

Reviewer #3:

The AAA+ protein Msp1 removes mislocalized tail-anchored (TA) proteins from the outer membrane of mitochondria. Here, the authors provide several cryo-EM structures of Msp1 at high resolution, allowing deducing the mechanisms of substrate engagement and ATP-hydrolysis driven substrate threading. In particular, the authors show (i) how substrates can laterally enter the translocation channel by partial Msp1 ring opening, (ii) tight interactions between the substrate and pore-1 and pore-2 loops, which are further stabilized by pore-3 loops and (iii) a novel structural element that allows for inter-subunit signaling and coordinated cycling of Msp1 subunits.

The presented study is very good; the structures provide important information on substrate engagement and threading by Msp1. The manuscript is very well written and figure presentations are excellent. I recommend publication in *eLife* after the authors have addressed the following points:

Major points:

– The N-terminal domain of Msp1 is composed of two helices and loops (α0/1 and L1/2) and mediates the selection and initial engagement of substrates. In the presented cryo-EM structures only a part of this domain is visible (α0 and L1), while the second part, which is most crucial for substrate binding (comprising α1 and L2, Li et al., 2019), is not. The authors speculate that differences in α0/L1 visibility reports on different accessibilities of the α1/L2 substrate binding sites. This statement seems problematic, as the crucial part of the substrate-binding site is not observed in the structure. The authors are therefore asked to better rationalize this conclusion or revise the respective statement.

– Results section: results from SEC runs should be provided as Supplementary Figure.

---

## [Author Response]

[Editors’ note: the authors resubmitted a revised version of the paper for consideration. What follows is the authors’ response to the first round of review.]

Reviewer #1:[…]Although there are a number of AAA family structures that have been solved by cryo-EM recently, this work stands out because Msp1 has a highly specialized function requiring a number of unique features. Like other members of the Meiotic-Clade, there is a wide seam in the Msp1 ring upon substrate binding that is seen in both katanin (Zehr et al., 2017) and spastin (Sandate et al., 2019). The authors propose that this hydrophobic seam is important for engaging mistargeted substrates that are improperly targeted to the mitochondria outer membrane. This is a very intriguing hypothesis and is consistent with previous work from this lab and others, but there is no testing of this hypothesis showing the role of this hydrophobic seam in engagement or extracting substrates. Ideally, some combination of mutagenesis in Msp1 or substrate would allow for experimental validation of this model for the hydrophobic seam.

The previously published study (Li et al., 2019) largely addressed the reviewer’s concerns by doing extensive mutagenesis on both Msp1 and the substrate. We apologize for the confusion caused by insufficient description of the published results. We now expanded our description in the Introduction.

Briefly, from the Msp1 side, Li et al. mutated a number of hydrophobic amino acids to alanine individually. Msp1 bearing these mutations displayed reduced pull-down of the substrate, increased level of mislocalized tail-anchored proteins, and failure to rescue yeast growth under the *get3* deletion background. Most of these mutations do not disrupt Msp1’s hexamer formation, indicating that they recruit the substrate in a hexameric context. From the side of the substrate (Pex15Δ30), several truncations showed that a minimum sequence (Δ1-311) consisting of the hydrophobic patch, transmembrane segment, and the tail at the extreme C-terminus is sufficient for interaction with and removal by Msp1. Further deleting the hydrophobic patch (Δ1-324) rendered the protein stable and unrecognized by Msp1. Mutating several hydrophobic amino acids in this patch to alanine abolished the substrate’s interaction with Msp1. Finally, insertion of the hydrophobic patch into a nonsubstrate (Gem1) transformed it into a Msp1 substrate. These results together strongly indicate that the hydrophobic interactions between the substrate and the N-domain of Msp1 are indeed responsible for initial substrate recruitment in a hexamer context.

Due to the lack of the hexamer structure, Li et al. could not assign the positions of these hydrophobic amino acids. In our work, we mapped these hydrophobic amino acids to the active hexamer (Figure 2). Due to the melting and refolding of α0 (described in Figure 2), these amino acids are only exposed in the bottom subunit (M6), and many of them are aligned next to the open seam. Our structures in combination with the rich biochemical evidence shown previously provide a convincing hypothesis that the substrate engagement could happen at the open seam.

The claim that the pore-loop 2 contacts with substrates are a completely new feature of this work is interesting. A number of studies have shown a role for pore loop 2 (or equivalent pore loops) in stabilizing engaged substrate, however this work reveals intimate direct contacts between sidechains in Msp1 with the backbone of putative substrates (specifically H227 as shown in Figures 3 D/F). This is intriguing in that these direct interactions have not been seen in other AAA structures – in fact, some AAA machines do not seem to care about the specific nature of the polypeptide backbones. For example, ClpX can translocate polymers with up to 10 methylene groups separating consecutive peptide bonds (Barkow et al., 2009) suggesting the specific placement of the carbonyls is not critical for substrate engagement. The current structures suggest more intimate contacts between Msp1 and its substrate that seem like they would constrain substrates more strictly. Ideally, determining whether this is actually the case experimentally would strengthen the impact of this almost exclusively structural study.

Thank you for your comment. As requested by the reviewer, we now made point mutations in pore-loop 2 and tested their effects on yeast growth. The results are shown in Figure 4. Briefly, several pore-loop 2 amino acids including R222, E226 and E228 are essential to Msp1’s normal function as mutating them severely inhibits growth. Similar to this work, the recently published high-resolution structures of spastin and katanin in complex with a poly glutamate substrate also showed close contacts of pore-loop 2 with the substrate. Our structure and the structures of spastin and katanin converge on the observation that pore-loop 2 of these meiotic clade proteins directly contact the substrate. The mutational analysis for all three proteins also supports the functional importance of this loop.

Reviewer #2:[…]Major points:1) The authors propose a mechanism of allostery that largely centers on the melting of a post-ISS loop they refer to as the NCL. The mechanistic details of this loop's melting and its relevance to intersubunit interactions are unclear. Can the authors define the interactions that the NCL that stabilize intersubunit interactions?

Thank you for your comment. We have added the description of these interactions to our Result section (subsection “Subunits along the spiral propagate a linear sequence of nucleotide states in the reaction cycle”). Briefly, at each ATP-bound intersubunit interface, the NCL stacks on top of L2 (amino acids 100-109, Figure 1D), the short linker that follows α1, which is a structural feature unique to the meiotic clade. The interactions between the two structural elements are mainly hydrophobic, that is, the close stacking between the backbones of the NCL and the hydrophobic side chains (such as Pro107 and Ile106 and Val101) is predicted to exclude water molecules and to achieve a gain in the entropy, both favoring this conformation.

Further, the quality of the cryo-EM density should enable the authors to specify the residue-specific responses to the repositioning of the Arginine fingers and the WD motif that results in its melting and should be described. Importantly, the functional relevance of these residues should be verified using the established yeast growth assay (or other biochemical assays).

As shown in Figure 4, at the ADP-bound interface, the maps of both the arginine fingers and the WD motif are of lesser quality compared to that of the ATP-bound interface, suggesting a loss of rigidity of these amino acids. However, the equilibrium positions of these amino acids (as shown by the map) do not change significantly between the ADP- and the ATP-bound states. Instead of capturing the amino acids repositioned in a different conformation, our map captured a state where they simply seem more flexible around the same equilibrium position. We have tested the mutations on the WD motif in the yeast growth assay. The results agree with this interpretation and are now included Figure 4.

Given that the NCL is highly variable in sequence and length among meiotic clade AAA+ proteins, the case for an NCL^-^based mechanism that is conserved across this subfamily is not particularly strong.

We agree with the reviewer that the NCL is highly variable in both sequence and length among the meiotic clade proteins. Therefore, what we observe in the Msp1 structures may not be conserved in the entire clade. However, as we pointed out, the interactions between the NCL and the L2 of the opposing subunit are mainly hydrophobic, so they do not require a particular amino acid sequence. Lengthwise, as long as the loop is long enough to touch the opposing subunit, the stacking interaction is entirely plausible. Although not every member of the meiotic clade has an NCL as long as that of Msp1, we did notice (Figure 1—figure supplement 1), that katanin’s NCL is even longer than that of Msp1 and therefore may play a similar role in intersubunit communication. In fact, the structure of the katanin open spiral showed density of a loop (the equivalent of the NCL in Msp1) stacking with the opposing subunit, although the use of the Walker B mutation prevented the observation of the nucleotide state-dependent conformation change of this loop. Thus, the NCL is a feature unique to Msp1/ATAD1 and perhaps katanin. We have expanded our Discussion on this point in our manuscript (Discussion paragraph four and six).

2) The authors identify a WD motif that is very likely to be involved in the mechanism of action of this and other meiotic clade AAA+ proteins. However, the D from this motif is also a conserved component of the ISS, and it was previously proposed in YME1 that the D of the ISS is involved in sensing the position of the consecutive arginines. Given that the W residue is not conserved across the meiotic AAA+ clade, the functional relevance of the W in this motif should be further explored through mutagenesis. How the corresponding residue might be relevant to function in other meiotic clade ATPases should also be speculated.

We made mutations to W220 and tested their effects on yeast growth. The results are shown in Figure 4. In brief, cells expressing Msp1W220A showed a mild and Msp1W220D a strong growth defect, indicating the importance of this amino acid. The W220 is strictly conserved in Msp1 (Figure 1—figure supplement 2). In the meiotic clade, this position is usually an aliphatic amino acid (L or M), although spastin has an aromatic amino acid (F), positioned to interact with the arginine fingers, although the function of this residue in the intersubunit interaction has yet to be tested. We have expanded our Discussion related to this residue in our manuscript (paragraph four).

3) As shown by the authors in Figure 1—figure supplements 1 and 2, in meiotic clade AAA+ proteins, pore loops 2 and 3 contain conserved positive and negatively charged residues that have been shown to be required for activity. The authors further describe a powerful interconnected network formed by positive and negatively charged residues in pore loops 2 and 3 (Figure 3—figure supplement 3), which is also seen in our structure of spastin (Sandate et al., 2019). In spastin, this network connects the pore loop of one subunit to the nucleotide-binding pocket of its counterclockwise neighbor, suggesting that this charge network might be the main driver of intersubunit communication and allosteric transmission of nucleotide state to the pore loops, which explains their essential role for activity. Could this also be the case in Msp1? Can the authors include the nucleotides in their current Figure 3—figure supplement 3? The authors should also evaluate the functional relevance of this pore loop charged network in Msp1 using their yeast growth assay (i.e. alanine substitutions of the conserved charged residues in the pore loops).

We observed an interconnected network that involves all three pore-loops (Figure 3C and Figure 3—figure supplement 2), where pore-loop 2 is sandwiched between pore-loops 1 and 3 of subunits on both sides. The unique positioning of pore-loop 2 enables it to sense the position of a subunit by sensing the presence of interaction partners on both sides. The reviewer raised an interesting point about the role of this charged network being an allosteric machinery that could transmit the nucleotide state from the enzyme’s pocket to the pore-loops. To test whether this could be the case for Msp1, we examined the conformations of pore-loop 3, which would be the connection between the nucleotide binding pocket and pore-loop 2 and 1 (directly interacting with the substrate) at both the ATP-bound and the ADP-bound intersubunit interface. Notably, we did not observe a significant difference (we added the nucleotide states in Figure 3—figure supplement 2 (previously Figure 3—figure supplement 3)), indicating that the state is insensitive to the nucleotide state. Rather, its conformations remain mostly unchanged across the ring, which is why we do not think that it could sense the nucleotide state change and transmit it to the substrate interacting pore-loops (pore-loops 1 and 2). Rather, we propose that it (together with pore-loop 1) stacks with pore-loop 2 (Figure 3C) across subunits, providing pore-loop 2 stabilization on both sides, providing a way to sense its position in the spiral. We performed extensive mutagenesis experiments on this charged network and showed that several mutations in this network inhibits Msp1’s function, namely, R222A, R222E, R264E and D269A. The results show that these amino acids are important to Msp1’s function, but most likely due to the aforementioned reason, and not the allosteric transmission.

4) Are changes in the nucleotide binding pocket being transmitted to the pore loops, or do the authors propose that the pore loops release from substrate as a result of the rigid-body displacement of the subunit as it transitions from the bottom to the top of the spiral?

In our structure, although we captured different nucleotide-bound states across the ring, we did not observe the melting and refolding of the α-helix that immediately proceeds the ISS as observed in the Yme1 structure. Therefore, we cannot conclude that there is allosteric transmission of the nucleotide states to the pore-loops. What we did observe, in terms of conformational changes between subunits, are (1) the melting of NCL in response to ATP hydrolysis (both by comparing the NCL of M5 to M6 in Δ30-Msp1 (closed), as well as the M6 in Δ30-Msp1 (closed) and Δ30-Msp1^E214Q^), and (2) retraction of pore-loop 2 of the bottom subunit (M6), which we believe could be an effect of the loss of stacking partner on the opposite side of the seam (Figure 3C). The structures suggest that these two elements are the first to initiate the upward translocation of M6. In that sense, the subunit does not move as a rigid body.

5) As discussed by the authors, the ISS motif (DGF) has emerged as an allosteric driver of inter-subunit communication in numerous AAA+ proteins. The authors claim that the substitution of DGF for DGL diminishes the role that the ISS motif plays in intersubunit coordination in Msp1, due to loss of π-stacking interactions with the core β-strands of the adjacent subunit. However, both F and L are hydrophobic residues, and the loop in Msp1 indeed appears to extend into the hydrophobic groove at the intersubunit interface where the L could engage in hydrophobic interactions (Figure 5—figure supplement 1). I agree that the absence of π-stacking interactions likely decreases the relevance of the ISS motif itself in the mechanism of allostery of Msp1, but the potential relevance of this loop to the Msp1 mechanism cannot be discounted. Further, ADP-BeF is known to mimic a transition state in AAA+ proteins, and it is thus reasonable to believe that the disordered, but unretracted ISS motif observed in the ADP-like subunit in fact corresponds to a transition state wherein the ISS motif is in a transitional refolding state. To gain a better understanding of this loop's functional role, the authors should incorporate the following:a) Evaluate the functional relevance of DGL in Msp1 using their yeast growth assay (or other biochemical assays) to determine the effect of DGL◊DGA (and ideally DGE and DGV mutations).

We have made mutations to L244 and tested their effects in the yeast growth assay (Figure 4). The results showed that L244A has a mild impact on Msp1’s function and that L244E strongly affects the protein’s function, which indicates that the hydrophobic interactions between L244 and the counter-clockwise adjacent subunit is important to Msp1’s function. We have expanded our discussion to include this point.

b) Expand their description of the nucleotide binding pocket of Msp1. Is the aromatic residue on the core β-strand that π-stacks with the ISS motif F in YME1/AFG3L2/Ftsh/26S proteasome not present in Msp1 and/or other meiotic clade AAA+ proteins? What interactions is DGL involved in, in addition to the D interaction with the consecutive arginines?

We have expanded our discussion to include the interactions of DGL. In brief, in YME1, a phenylalanine in the ISS motif stacks with three phenylalanines in the opposing subunit. By contrast, in Msp1, the phenylalanine is replaced with a leucine (L244) and forms hydrophobic stacking with two phenylalanine residues (F211 and F175) in the core β-strands in the counter-clockwise adjacent subunit. An asparagine (N177) replaces the third phenylalanine in YME1 and does not appear to form interactions with L244.

c) Include a figure showing the EM density for the ISS motif in the apo subunit, and try to assess whether it undergoes a refolding event. I understand that the quality of the reconstruction for this mobile subunit is not sufficient for modeling, but the formation of a secondary element might be visible even at resolutions close to 8 Å.

We have included a figure (Figure 6—figure supplement 2) showing the EM density of the aposubunit as requested by the reviewer. Unfortunately, due to the dynamic nature of this subunit, the map quality is significantly worse than those of other subunits. With the current map, we cannot assess with confidence whether a refolding event has taken place in the mobile subunit.

6) I agree with the authors' overarching conclusion that Msp1 has evolved to function as a more powerful unfoldase through: i) double aromatic in pore loop 1, ii) substrate interacting pore loop 2, iii) a unique mechanism of allostery. This is the most important finding of this study, and I encourage the authors to expand their conclusion to include similar observations across the AAA+ superfamily:a) The degenerate ISS motif and the consequences for the mechanism of allostery are a very important part of the mechanism of Msp1. In addition to Type I AAA+ proteins of the meiotic clade, degenerate ISS motifs are also present in other classical AAA+ proteins, such as Type II AAA+ protein NSF (DGV instead of DGF, White et al., 2018) and AAA+ protease Paraplegin (DGM instead of DGF, Figure 1—figure supplement 1). Can the authors discuss their findings in the context of the role of degenerate ISS motifs across the AAA+ superfamily?

We have expanded our Discussion to include other proteins that have similar degenerated ISS motifs (paragraph six).

b) Increased bulkiness of the pore loops was shown to affect both the chemical and mechanical properties of the AAA+ motor (Rodriguez-Aliaga et al., 2016). Can the authors discuss their findings regarding the pore loops in this context?c) The presence of two aromatic residues in pore loop 1 is noteworthy, and the authors should reference the recent papers of Type II AAA+ protein Cdc48 bound to substrate (Cooney et al., 2019, Twomey et al., 2019) that showed this same organization, and discuss how distantly related AAA+ proteins have converged on similar solutions to increase their grip on substrate.d) Similarly, pore loop 2 has recently been shown to directly contact the substrate in the mitochondrial inner membrane AAA+ protease AFG3L2, which also appears to have evolved to be a more powerful unfoldase (Puchades et al., 2019). An expanded discussion on pore loop 2 interactions in different meiotic clade AAA+ proteins, as well as distantly related AAA+ proteases, to those in Msp1 would increase the impact of the authors' findings.

We have expanded our Discussion to reflect all three points (paragraph eight).

7) In Figure 1—figure supplement 5, the authors discuss the presence of heptamers in their sample, which might be an artifact of particles being aligned with an offset of 1 subunit register relative to one another. Have the authors attempted to further classify this heptameric class into subsets? In our recent work with the closely related Type I AAA+ protein spastin (Sandate et al., 2019), we also initially observed a heptameric reconstruction, but found that this organization was a processing artifact (described in the Materials and methods). Given that both spastin and Msp1 are single-ring Type I ATPases of the meiotic clade, and the spastin "heptamer" we observed contained a Walker B mutation, the authors may be encountering the same register misalignment.

We considered the possibility of register misalignment could be the cause of the observed heptamer species. Therefore, we further classified the heptamer class and observed that a subset of the particles displayed extra density for an additional subunit (an octamer class), but not a subclass of hexamers, suggesting true continuous growth along the spiral in some particles. We do not think the observed continuous growth is due to the misalignment in 3D, because when we did 2D classification of the heptamer class, we observed side views with two layers of subunits, showing the presence of additional subunits (Figure 1—figure supplement 6). By contrast, the 2D classification of the hexamer class showed side views that display only a single layer of subunits, corresponding to the structure presented in the paper. This clearly rules out the possibility that the extra density observed in the heptamer class results from misalignment in 3D.

Reviewer #3:[…]Major points:– The N-terminal domain of Msp1 is composed of two helices and loops (α0/1 and L1/2) and mediates the selection and initial engagement of substrates. In the presented cryo-EM structures only a part of this domain is visible (α0 and L1), while the second part, which is most crucial for substrate binding (comprising α1 and L2, Li et al., 2019), is not. The authors speculate that differences in α0/L1 visibility reports on different accessibilities of the α1/L2 substrate binding sites. This statement seems problematic, as the crucial part of the substrate-binding site is not observed in the structure. The authors are therefore asked to better rationalize this conclusion or revise the respective statement.

Thank you for pointing out that part of the LD is invisible in our structure. However, the reviewer was mistaken about which part is invisible: in the Δ30-Msp1closed structure, we were able to model the majority of the LD with the exception of residues 65-85, which constitutes part of L1. The entirety of α1 (residues 90-97) and L2 (residues 97-103) were modeled in the structure. The reviewer is correct that most of the residues shown to be important in substrate recruitment are located in the α1/ L2 region, and they are indeed modeled in our current structure (see Figure 2D and E). It is important to note that *C.t.* Msp1 has a much longer L1 than *S.c.* Msp1 (see sequence alignment in Figure 1—figure supplement 2) and that half of the residues that are invisible in our structure actually belong to the longer part that is not conserved in the *S.c.* Msp1, in which the hydrophobic patch was initially discovered. Therefore, our hypothesis that the melting and refolding of α0 affects the accessibility of the substrate recruitment site remains valid.

– Results section: results from SEC runs should be provided as Supplementary Figure.

We have added a figure supplement to show the results of the SEC runs (Figure 1—figure supplement 3).